# Reduction of chromosomal instability and inflammation is a common aspect of adaptation to aneuploidy

Dorine C Hintzen [ID][1], Michael Schubert [ID][1], Mar Soto[1], René H Medema [ID][1,2✉] & Jonne A Raaijmakers [ID][1✉]

## Abstract

**Aneuploidy, while detrimental to untransformed cells, is notably prevalent in cancer. Aneuploidy is found as an early event during tumorigenesis which indicates that cancer cells have the ability to surmount the initial stress responses associated with aneuploidy, enabling rapid proliferation despite aberrant karyotypes. To generate more insight into key cellular processes and requirements underlying adaptation to aneuploidy, we generated a panel of aneuploid clones in p53-deficient RPE-1 cells and studied their behavior over time. As expected, de novo-generated aneuploid clones initially display reduced fitness, enhanced levels of chromosomal instability (CIN), and an upregulated inflammatory response. Intriguingly, after prolonged culturing, aneuploid clones exhibit increased proliferation rates while maintaining aberrant karyotypes, indicative of an adaptive response to the aneuploid state. Interestingly, all adapted clones display reduced CIN and reduced inflammatory signaling, suggesting that these are common aspects of adaptation to aneuploidy. Collectively, our data suggests that CIN and concomitant inflammation are key processes that require correction to allow for fast proliferation in vitro. Finally, we provide evidence that amplification of oncogenic KRAS can promote adaptation.**

**Keywords** Aneuploidy; Adaptation; Chromosomal Instability; Inflammation
**Subject Categories** Cancer; Cell Cycle

## Introduction

Each time a cell divides, it must distribute its genomic content equally amongst daughter cells. Cell division is a highly regulated process, safeguarded by various checkpoints such as the spindle assembly checkpoint (Kops et al, 2005). However, when errors occur, chromosomes can missegregate, giving rise to cells with erroneous karyotypes. The state of having an aberrant number of chromosomes is known as aneuploidy.

Aneuploidy manifests deleterious effects at both the organismal and cellular levels, resulting in developmental defects and reduced cellular proliferation (Ariyoshi et al, 2016; Bonney et al, 2015; Santaguida and Amon, 2015b; Sheltzer et al, 2017; Stingele et al, 2012; Torres et al, 2007; Williams et al, 2008). The detrimental effects of aneuploidy have been attributed to imbalances in coding genetic material (Torres, 2023; Torres et al, 2007). It is widely acknowledged that aneuploidy results in the altered expression of the genes located on the involved chromosome(s). While RNA expression largely scales with DNA copy number (Dephoure et al, 2014; Pavelka et al, 2010; Schukken and Sheltzer, 2022; Stingele et al, 2012; Torres et al, 2007, 2010; Upender et al, 2004; Williams et al, 2008), large-scale dosage compensation is observed at the protein level in an attempt to maintain complex stoichiometry and appropriate levels of dosage sensitive factors (Dephoure et al, 2014; Hwang et al, 2021; McShane et al, 2016; Pavelka et al, 2010; Senger et al, 2022; Stingele et al, 2012; Viganó et al, 2018). The disruption of proteostasis upon aneuploidy results in an increased burden on protein synthesis, folding, and degradation machineries, often leading to proteotoxic stress (Deshaies, 2014). Indeed, in both aneuploid yeast and human cell models, indications of proteotoxic stress and a higher dependency on processes related to the maintenance of proteostasis have been extensively documented (Ariyoshi et al, 2016; Brennan et al, 2019; Dephoure et al, 2014; Donnelly et al, 2014; Hanna et al, 2006; Ohashi et al, 2015; Oromendia et al, 2012; Santaguida et al, 2015; Santaguida and Amon, 2015a; Stingele et al, 2012).

Aneuploidy has also been linked to enhanced genetic instability (Sheltzer et al, 2011; Torres et al, 2007) and replication stress (Blank et al, 2015; Garribba et al, 2023; Ohashi et al, 2015; Passerini et al, 2016; Santaguida et al, 2017; Torres et al, 2007), which can be caused by imbalances in crucial protein complexes (Passerini et al, 2016). Besides, numerous studies have reported metabolic alterations in response to aneuploidy (Foijer et al, 2014; Hwang et al, 2017; Sheltzer, 2013; Stingele et al, 2012; Tang et al, 2017; Torres et al, 2007). These metabolic alterations might be caused by the enhanced energy requirements to deal with the challenges induced by the aneuploid state. Importantly, the maintenance of metabolic

[1]Oncode Institute, Division of Cell Biology, The Netherlands Cancer Institute, Plesmanlaan 121, 1066 CX Amsterdam, The Netherlands. [2]Present address: Oncode Institute, Princess Maxima Center for Pediatric Oncology, Heidelberglaan 25, 3584 CS Utrecht, The Netherlands. ✉E-mail: r.h.medema-2@prinsesmaximacentrum.nl; j.raaijmakers@nki.nl

homeostasis is also contingent on the stoichiometry of metabolic regulators and enzymes, making it susceptible to the effects of gene imbalances. Moreover, elevated levels of reactive oxygen species (ROS) are present in aneuploid cells (Ariyoshi et al, 2016; Dephoure et al, 2014; Li et al, 2010), which poses a further threat to genomic stability. Altogether, aneuploidy has a severe impact on the proteostasis of the cell, thereby disrupting many cellular processes, inducing stress responses and forming a major threat to the fitness and genomic stability of the cells.

While aneuploidy arises as a consequence of errors during mitosis, aneuploidy has also been shown to be a driver of ongoing mitotic errors, a phenomenon termed chromosomal instability (CIN) (Duesberg et al, 1998; Garribba et al, 2023; Nicholson et al, 2015; Passerini et al, 2016; Sheltzer et al, 2011; Thompson and Compton, 2010; Zhu et al, 2012). The induction of CIN has also been linked to gene imbalances (Sheltzer et al, 2011; Zhu et al, 2012), and recent work shows that CIN levels specifically correlate to the amount of gained coding genes (Hintzen et al, 2022). Continuous CIN can evoke an inflammatory response via the cGAS-STING pathway as a consequence of genetic material that ends up in the cytoplasm due to rupture of micronuclei or chromatin bridges that persist (Dou et al, 2017; Flynn et al, 2021; MacKenzie et al, 2017; Sun et al, 2013). In addition, CIN can give rise to cellular populations harboring complex karyotypes, capable of inducing inflammation through entering a senescent-like state and the acquisition of a senescence-associated secretory phenotype (SASP) (Santaguida et al, 2017; Wang et al, 2021).

In light of the extent of stress responses elicited by aneuploidy, it is apparent that aneuploidy represents a highly detrimental condition. At first sight it therefore seems paradoxical that aneuploidy is highly prevalent in cancer, a disease characterized by fast and uncontrolled proliferation, with ~90% of all solid tumors being aneuploid (Taylor et al, 2018; Storchova and Kuffer, 2008; Vasudevan et al, 2021; Weaver et al, 2007). This phenomenon is therefore referred to as the "aneuploidy paradox" (Sheltzer and Amon, 2011; Weaver and Cleveland, 2008). Many studies have described the positive effects of aneuploidy and CIN on tumor heterogeneity, tumor progression and its contribution to therapy resistance (Andrade et al, 2023; Lukow et al, 2021; Ippolito et al, 2021; Shoshani et al, 2021; Trakala et al, 2021), highlighting the benefits for tumors to induce and maintain aneuploidy. However, the adaptive mechanisms of cells to aberrant karyotypes are currently poorly understood.

One important factor in aneuploidy tolerance in human cells is p53 (Marques and Kops, 2023; Soto et al, 2017). However, even in the context of p53-deficiency, de novo aneuploidies severely impact cellular fitness (Chunduri et al, 2021; Hintzen et al, 2022; Passerini et al, 2016; Sheltzer et al, 2011; Taylor et al, 2018). A few tolerance mechanisms have been shown to be at play independently of p53. For example, in yeast, that do not carry a TP53 gene, adaptation to aneuploidy has been shown to involve mutations in genes involved in mTOR signaling (Kaya et al, 2020) or in protein degradation (Torres et al, 2010). Initial tolerance to de novo aneuploidies in human cells has been shown to be determined by p38-mediated metabolic alterations (Simões-Sousa et al, 2018). Besides, loss of BRG1, a member of the SWI/SNF complex, has been shown to alter initial aneuploidy tolerance, possibly also by affecting the metabolic state of the cell (Schiavoni et al, 2022). However, less is understood about the processes underlying long-term adaptation to aneuploidy

in human cells. One study shows that restoring Hsp90-mediated protein folding promotes adaptation to aneuploidy in cell culture (Donnelly et al, 2014). However, the exact adaptive mechanisms that are employed by aneuploid cells are currently not clear (Hintzen, 2024).

In this study, we set out to elucidate the general requirements and mechanisms via which human cells overcome the stress responses driven by aneuploidy over a prolonged period of time. For this, we utilized RPE-1 p53kd cells, a near-diploid p53-deficient human cell line. With this model system, we aimed to uncover fundamental principles of how cells adapt to de novo aneuploidy induction in vitro. To exclude karyotype-specific adaptation pathways, we made use of a large panel of different aneuploid clones with a variety of aneuploidy levels. Using this system, we systematically uncovered commonalities intrinsic to cells adapting to aneuploidy.

## Results

### Cells can adapt to aneuploidy

To investigate mechanisms of adaptation to aneuploidy, we generated a panel of aneuploid clones originating from an hTERT-immortalized retinal pigment epithelial cell line (RPE-1). Notably, given the pivotal role of p53 in aneuploidy tolerance, we conducted these experiments using clones harboring a stable shRNA-mediated knockdown of p53 (Soto et al, 2017). The generation of de novo aneuploid clones was described previously (Hintzen et al, 2022). In short: we induced random aneuploidies by 24-h treatment with low doses of MPS1 and CENP-E inhibitors after which cells were single-cell plated and grown until clones were established (Fig. 1A). Aneuploid clones were identified by determining large copy number variations (CNV) by low-pass whole-genome sequencing (raw reads have been deposited in NCBI's Sequence Reads Archive (SRA) under number SRP511546). We selected 14 clones with a variety of aneuploidy levels, mostly involving entire chromosomes (Appendix Fig. S1A). We cultured these clones for a time span of several months. To account for potential effects due to extended culturing as well as aging, we subjected the parental cells to the same prolonged culturing conditions. At monthly intervals, we measured the doubling times of these clones using live-cell imaging as a read-out for cellular fitness and, by extension, adaptation (see Fig. 1A for the full working pipeline). As expected, and in line with our previous report (Hintzen et al, 2022), all clones initially displayed a proliferation defect, albeit to different extents. After being subjected to long-term culturing, the majority of clones decreased their doubling times almost completely to that of parental RPE-1 cells, whilst a subset of clones (12.10, 14.17, 14.20) remained impaired in their proliferation as compared to parental cells. However, remarkably, all selected clones exhibited a prominent increase in proliferative capacity over time, indicative of adaptation (Fig. 1B).

### Full karyotype correction is not a common mechanism of adaptation

To gain insights into the adaptation process of the adapted clones, we conducted RNA sequencing on all 14 early aneuploid clones and

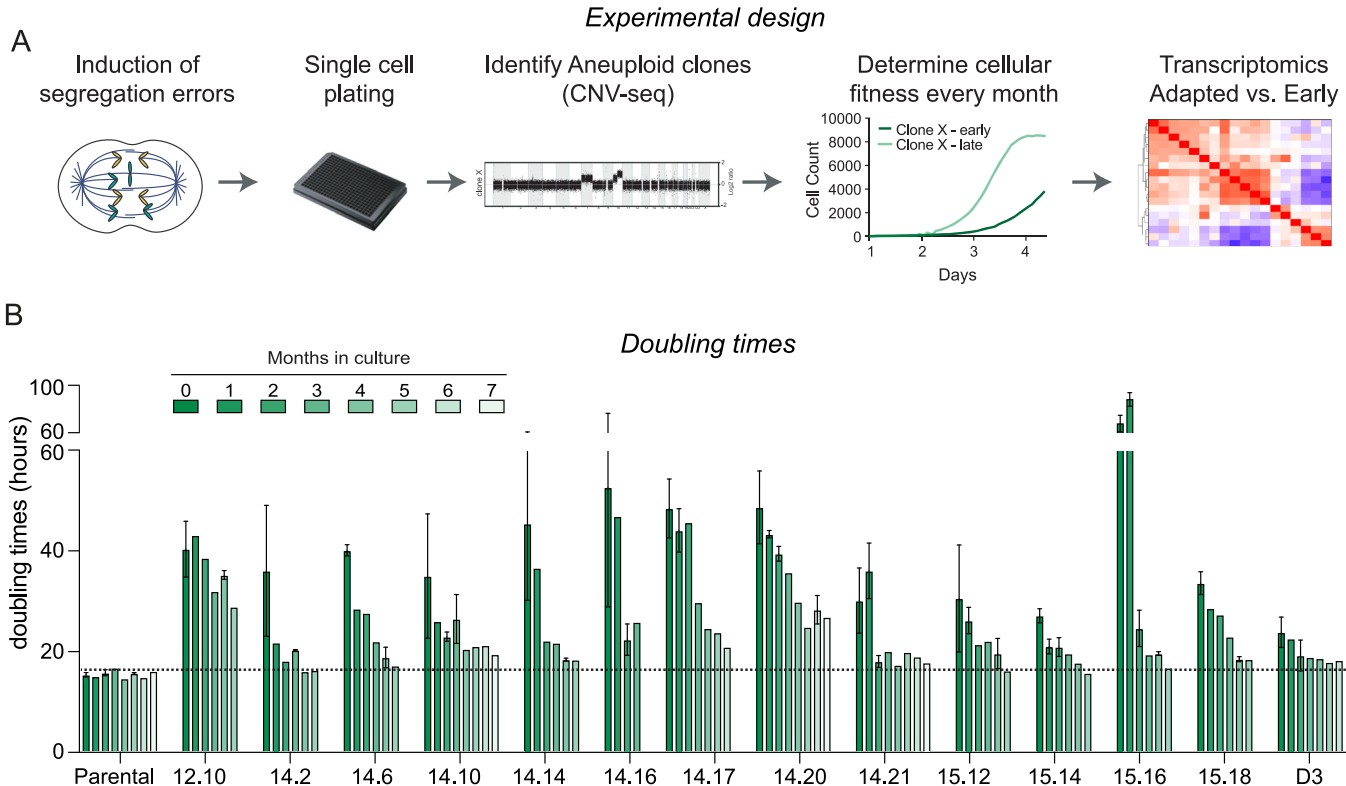

**Figure 1. Cells adapt to aneuploidy.**

(A) Working pipeline. Parental RPE-1 p53KD cells were treated with MPS1 and CENP-E inhibitors to induce segregation errors and plated single-cell to generate clones. The ploidy of the clones was determined using low-pass whole-genome sequencing and only aneuploid clones were kept for further investigation. Genome sequencing data is available from NCBI's Sequence Read Archive (SRA) under number SRP511546. The aneuploid clones were subjected to long-term culturing, and adaptation was monitored using live-cell imaging. To elucidate adaptation mechanisms, RNA sequencing was performed. (B) Doubling times of parental RPE-1 p53KD, and aneuploid clones determined via live-cell imaging performed at monthly intervals. Bars show mean doubling times, error bars indicate standard deviation between experiments. $n =$ at least three experiments. In-between time points were only imaged once and hence do not contain error bars. Source data are available online for this figure.

their adapted counterparts (raw transcriptome data has been deposited in NCBI's Gene Expression Omnibus (GEO) under number GSE273576). For the adapted clones, we selected the first time point at which they proliferated at their maximum capacity, minimizing the potential confounding effects of aging. The most obvious path to adaptation would be karyotype correction, where aneuploid clones revert to a (more) euploid karyotype. Since RNA levels largely scale with DNA copy number (Dephoure et al, 2014; Pavelka et al, 2010; Schukken and Sheltzer, 2022; Stingele et al, 2012; Torres et al, 2007, 2010; Upender et al, 2004; Williams et al, 2008), we first utilized the transcriptome data to extract karyotype information (Fig. 2A: red regions reflect gains, blue regions reflect losses). This method confirmed the karyotypes of the early clones determined with low-pass whole-genome DNA sequencing, validating the reliability of using RNA-seq reads for deducing karyotypes (compare Appendix Fig. S1A and Fig. 2A). For four clones, we determined that upon adaptation, the RNAseq reads still reflect those of DNA copy number (Appendix Fig. S1B). This not only shows that the deduction of karyotypes can be reliably performed both in early and adapted clones but it also suggests that large-scale dosage compensation on the level of the transcriptome

is unlikely to contribute to adaptation. When we examined the karyotypes of early clones in comparison to their adapted counterparts, we observed distinct patterns of karyotype alterations. Importantly, although some clones displayed a relatively stable karyotype upon adaptation, most clones displayed some levels of karyotype alteration. To obtain a quantitative measurement of these alterations, we determined the number of imbalanced genes for each clone by applying cutoffs on the fold change values of gene expression (see methods). Using these measurements, we categorized the karyotype progression upon adaptation into four distinct groups: "more complex" (karyotypes obtain more imbalanced genes), "evolved" (karyotypes changed but maintained a highly aneuploid karyotype), "simplified" (remains aneuploid, but less complex karyotypes), and "reverted" (karyotype similar to the parental RPE-1).

Out of 14 clones analyzed, a subset of clones displayed an "evolved" karyotype (14.3%) or became even "more complex" (21.4%) in terms of their karyotypes. However, the majority of clones exhibited karyotype simplification (57.1%). Notably, only one clone (14.14) fully reverted its karyotype to the parental karyotype, emphasizing that complete karyotype correction is not a

A    *Copy numbers from RNA−seq*

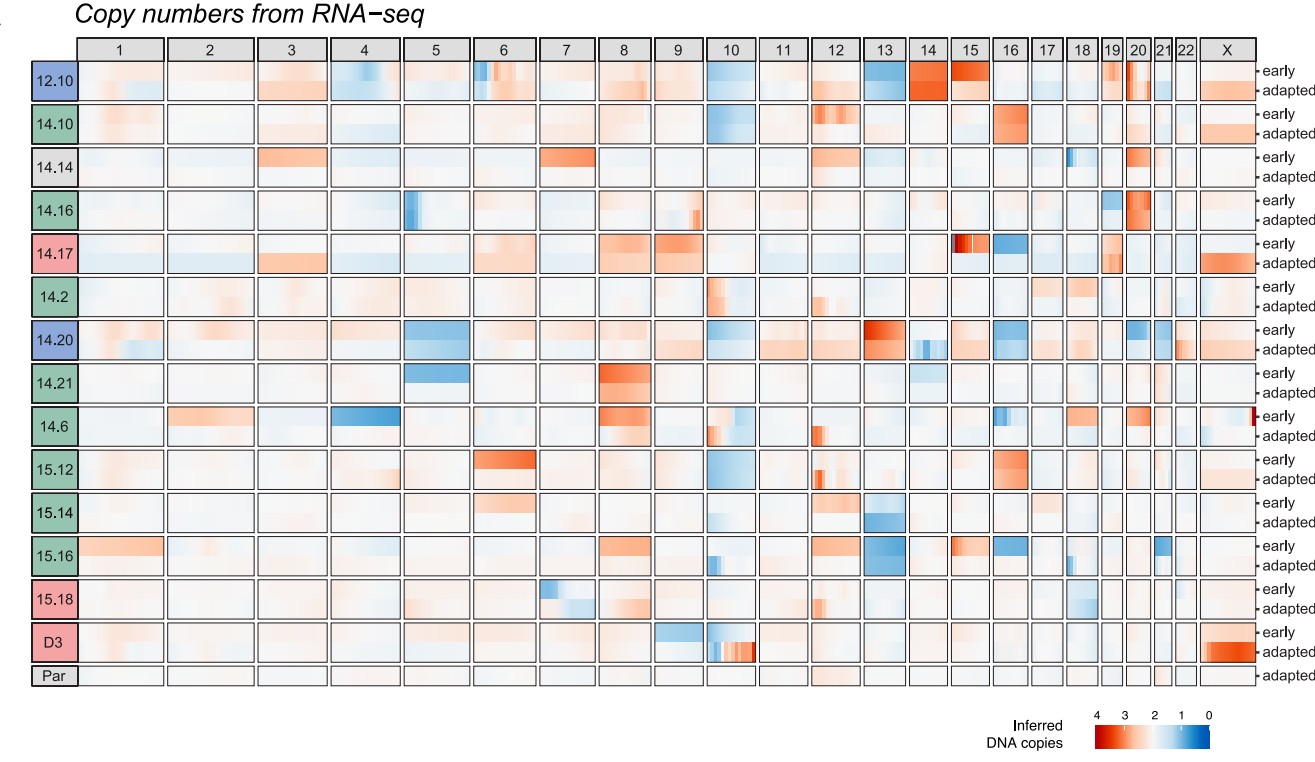

B    *Imbalanced Genes*

C    *Proliferation Rates vs. Imbalanced Genes*

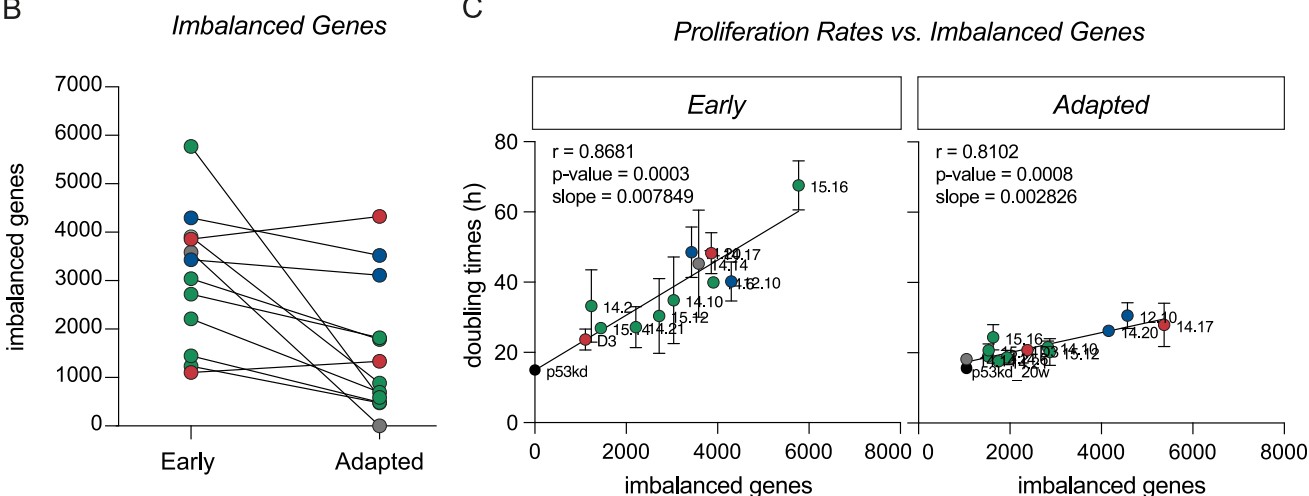

**Figure 2.  Full karyotype correction is not a common mechanism of adaptation.**

(A) Karyotypes of early and adapted clones derived from RNA-sequencing data, using RPE-1 parental p53KD cells as a reference. Clones are color-coded by their adaptation behavior: blue clones are clones with an "evolved" karyotype, green clones belong to the "simplified" karyotypes class, gray clones refer to clones with a completely "reverted" karyotype and red clones obtained "more complex" karyotypes. (B) The number of imbalanced genes per clone calculated from RNA-sequencing data in early and adapted clones. Lines connect early clones to their adapted counterparts. Color-coding as determined in (A). A paired *T* test was performed between early and adapted clones, *P* = 0.0109. (C) Spearman correlation between total coding-gene imbalances per clone and proliferation rates measured in (B) for early and adapted clones. Dots represent mean, error bars indicate standard deviation, *n* = 2. Line shows linear regression model fitted to the data. Color-coding as determined in (A). Source data are available online for this figure.

common mechanism of adaptation. Importantly, when we compared the number of imbalanced genes of all early clones to their adapted counterparts, we observed that the extent of karyotype simplification was overall significant (*P* = 0.0109, paired *t* test), but relatively modest for most clones (Fig. 2B). This suggests

that the observed karyotype simplification cannot fully account for the enhanced proliferation rates, that often reaches the rates of parental RPE-1 cells.

It has been shown by us and others that proliferation rates strongly correlate to the number of imbalanced genes (Dephoure

et al, 2014; Hintzen et al, 2022; Pavelka et al, 2010; Stingele et al, 2012; Torres et al, 2007). If adaptation to aneuploidy is indeed not solely driven by karyotype simplification but other processes contribute to the adaptation process, we predicted that the impact of imbalanced genes on proliferation rates will be less pronounced in adapted clones as compared to their early counterparts. Consistent with our earlier investigation, we observed that imbalanced genes exhibit a strong correlation with doubling times in early clones (Fig. 2C, $r = 0.8681$, $P = 0.0003$), suggesting that the extent of gene imbalances indeed determines the extent of the proliferation defect, at least in the early clones. In line with our previous observations, the number of imbalanced genes explained the proliferation defect better as compared to the number of gained or lost genes separately (Appendix Fig. S1C). Importantly, upon adaptation, we observed a strong and significant reduction in the slope of the linear regression model applied to imbalanced genes' data in the adapted clones (Fig. 2C, shifting from 0.007849 to 0.002826, $P = 0.0337$), in agreement with our prediction. This suggests that the impact of gene imbalances on doubling time reduces during the adaptation process, resulting in enhanced proliferation rates. It is important to note that the slope does not reach 0, underscoring that even in adapted clones, the presence of imbalanced genes continues to exert an effect on proliferation, albeit to a lesser degree compared to early clones. Altogether these data show that karyotype simplification or correction is not a common driver of adaptation but that active adaptation mechanisms must be at play.

## Reduced CIN and inflammation are common aspects of adaptation

Adaptation may involve the development of mechanisms to downregulate the initial stress responses triggered by aneuploidy or to improve the ability to tolerate these stress responses. To gain insight into the underlying mechanisms through which clones adapt to aneuploidy, we performed RNA sequencing and executed gene set enrichment analysis (GSEA) to assess which hallmarks and which biological processes were mostly altered during adaptation (Fig. 3A; Appendix Fig. S2A). Hallmarks that were most prominently upregulated upon adaptation were hallmarks associated with cell cycle, such as E2F targets and G2-M checkpoint (Fig. 3A,B), which is expected, as the adapted clones have increased proliferation rates (Fig. 1B). Similarly, predominantly upregulated biological processes are also attributable to the increased cellular such as translation, ribosome biogenesis, mRNA processing, and DNA replication (Appendix Fig. S2A). Hallmarks that were most prominently downregulated upon adaptation were hallmarks associated with inflammatory responses such as TNFα signaling via NF-κB and IL6/JAK/STAT3 (Fig. 3A,B). Importantly, these alterations were reproducibly observed in the majority of individual clones (Fig. 3B; Appendix Fig. S2B–D), suggesting that these are uniform responses in adaptation. Importantly, most alterations were in fact corrections of the early response to aneuploidy (Fig. 3A; Appendix Fig. S2A, see middle row). Moreover, GSVA scores suggest that the responses upon adaptation are overall not normalized completely to parental levels, suggesting that despite normalized proliferation rates, aneuploid clones remain altered in their transcriptomes (Fig. 3B; Appendix Fig. S2D). To search for alterations that might actively drive adaptation, we analyzed

gene set alterations that were unique to the adapted clones, and were unaltered in early clones. Strikingly, we only found peroxisomes and glycolysis to be specifically upregulated (Fig. 3C), suggesting that adapted clones might undergo metabolic rewiring.

It is known that aneuploidy itself can drive CIN and CIN is an established instigator of inflammation. Therefore, we hypothesized that the reduction in inflammation might be a consequence of decreased aneuploidy-induced CIN during the process of adaptation. To test this, we conducted live-cell imaging on both early aneuploid clones and adapted clones and quantified the frequency of errors that occurred during anaphase. As expected, we observed that all early clones showed elevated CIN levels compared to parental cells (Fig. 4A). Interestingly, adapted clones all displayed lower levels of CIN when compared to their early counterparts, suggesting that downregulation of CIN is a common aspect of adaptation. Indeed, we find that the reduced CIN rates strongly correlate to the improved proliferation rates upon full adaptation (Fig. 4B). To test if CIN levels follow the improved proliferation rates also during the adaptation process, we monitored missegregation rates over the time course of the adaptation process for several clones (Fig. EV1A). We observed that CIN levels consistently decreased during the adaptation trajectory in all clones, albeit with different kinetics. Overall, the decrease in CIN resembled the adaptation behavior in terms of doubling times which we exemplified by plotting doubling times against CIN rates, where both CIN levels and doubling times decrease simultaneously during adaptation (Fig. 4C,D) (Hintzen, 2024).

In our previous research, we demonstrated that the gained genetic material is the main determinant of CIN (Hintzen et al, 2022). In accordance, we observed the strongest correlation between gained genes and missegregation rates in early clones in this study ($r = 0.7978$, $P = 0.0016$) (Figs. 4E and EV1B). Interestingly, upon adaptation, we observed a profound decrease in the slope of the correlation between gained genes and CIN levels (0.008766–0.002499, $P < 0.001$), suggesting that gained genetic material has a dramatically lower impact on CIN levels in adapted clones as compared to early clones.

We showed before that proteotoxic stress can be a driver of CIN (Hintzen et al, 2022). We found that the early aneuploid clones express higher levels of the two key players downstream of PERK activation, namely ATF3 and CHOP/GADD153, indicative of an unfolded protein response (UPR) in the early clones (Read and Schröder, 2021) (Fig. EV2A). Upon adaptation, both ATF3 and CHOP were significantly downregulated, nearly reaching the levels of parental cells (Fig. EV2A), suggesting that proteotoxic stress is no longer induced, despite the presence of abnormal karyotypes. We selected three early clones with the highest expression of ATF3 for further characterization (Fig. EV2B). To investigate the role of PERK-signaling on proliferation, we treated cells with an inhibitor against PERK, which showed to effectively suppress the elevated ATF3 expression (Fig. EV2C). However, inhibition of PERK-signaling did not result in enhanced proliferation in the tested early clones (Fig. EV2D), suggesting that the PERK-mediated UPR signaling is unlikely solely responsible for the observed proliferation defects and CIN in early aneuploid clones. Thus, although correction of PERK-mediated UPR characterizes adaptation, suppression of of this response alone is insufficient to drive adaptation in early clones.

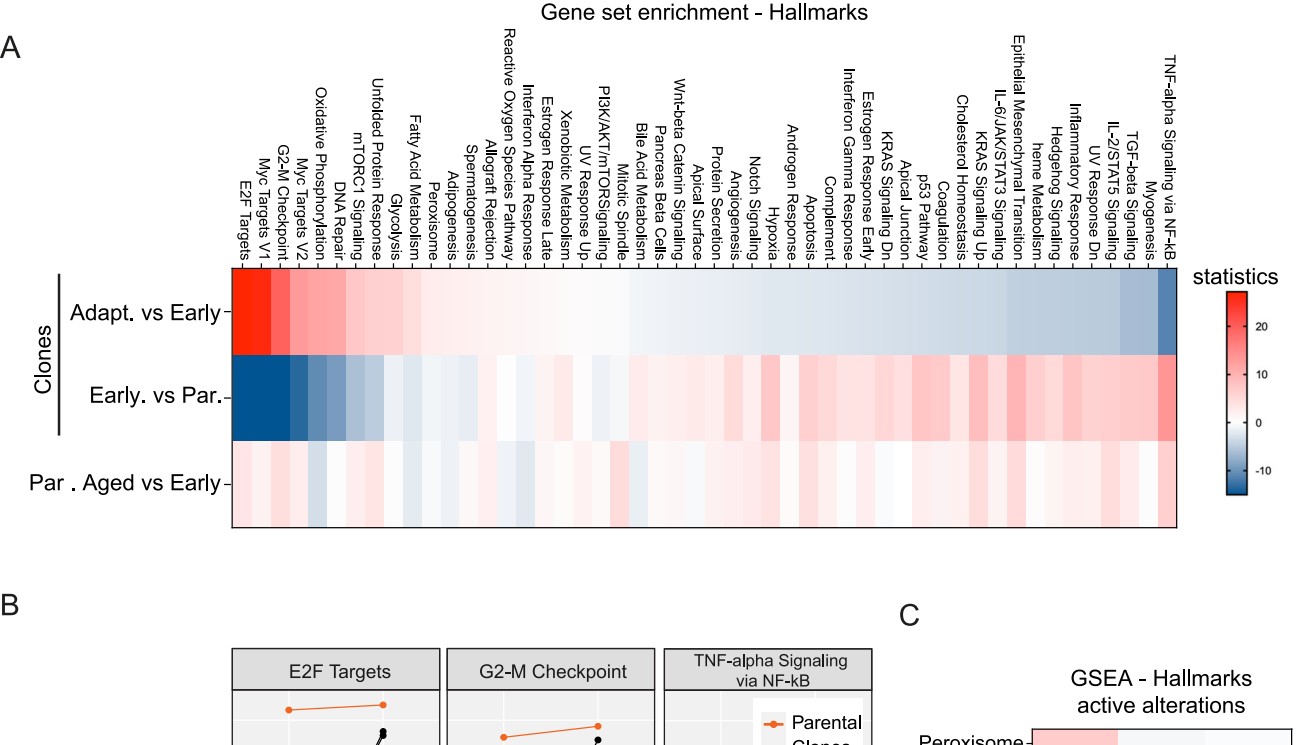

**Figure 3. Reduction of inflammation and upregulation of cell cycle hallmarks are common aspects of adaptation.**

(A) Gene set enrichment for hallmarks based on the transcriptional alterations observed in adapted clones compared to their early counterparts, colors indicate Wald statistics. The top 50 most altered gene sets in the adapted clones are displayed (top row). Alterations in early clones over parental (middle row) and aged parentals (bottom row) are displayed as a reference. (B) Plots showing Gene Set Variation Analysis (GSVA) for a selected number of hallmarks that show high alteration upon adaptation for the individual clones. Lines connect early clones with their adapted counterparts. The parental cells are displayed in orange as a reference. (C) Selected hallmarks (A) that showed alterations upon adaptation but that were minimally altered in early clones versus parental cells, indicative of pathway activation rather than correction. Source data are available online for this figure.

## Multiple consequences of CIN contribute to inflammation

Our study shows that aneuploidy triggers CIN and inflammation in early clones. Since both CIN and inflammation are corrected simultaneously upon adaptation, we speculate that these are directly linked and their correction might be required for adaptation. To uncover potential drivers of inflammation in the early clones, we performed GSEA on the transcriptome data. Gene sets for STING signaling, NF-κB activation and SASP were all positively enriched in the early clones (Fig. EV3A), indicating that multiple inflammatory pathways are activated. To investigate if the inflammatory response is a direct consequence of CIN, we aimed to compare the inflammatory response in our early clones to that of parental cells with induced missegregations. To this end, we treated

RPE-1 p53KD parental cells with CENP-E and MPS1 inhibitors for 48 h to induce missegregations and performed transcriptomics. Strikingly, we found very similar enrichment scores for the different inflammatory gene sets upon acute CIN compared to early clones (Fig. EV3A). This suggests that the inflammatory response in early aneuploid clones can be largely explained by the enhanced CIN levels. Moreover, it shows that this inflammatory response is complex and involves multiple pathways. To explore the impact of an upregulated inflammatory response on cell proliferation, we utilized synthetic cGAMP to mimic the activation of the STING pathway in non-aneuploid parental cells. Although earlier reports described the absence of cGAS and STING from RPE-1 cells (Basit et al, 2020), our transcriptome data suggests prominent expression of STING while cGAS expression was low. Most

A

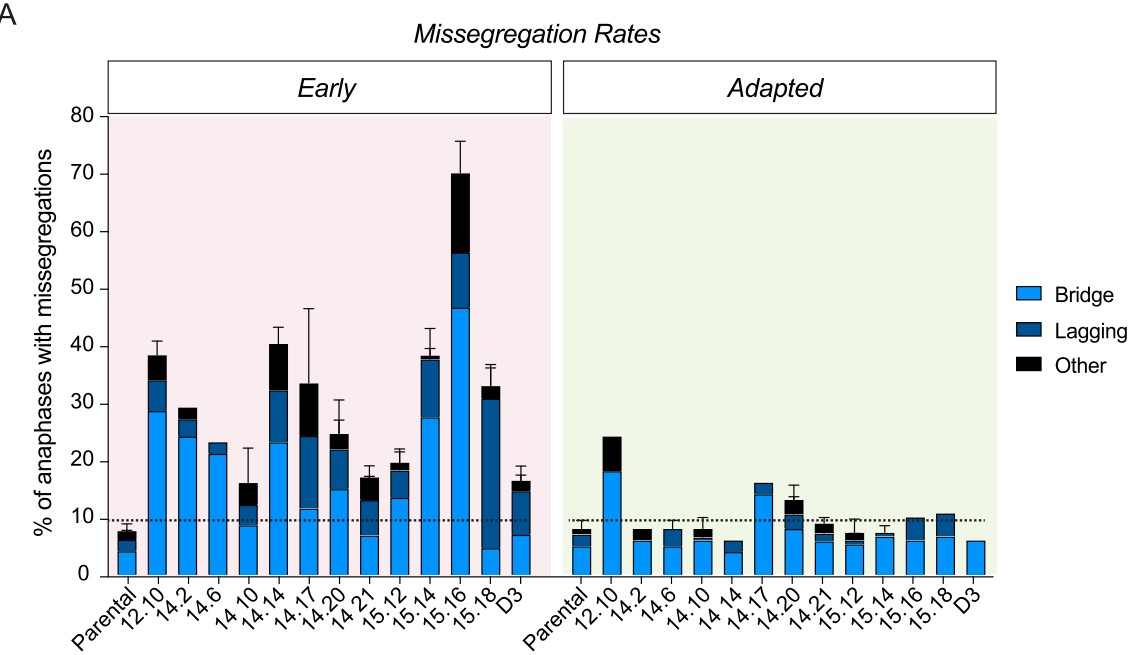

B

C

D

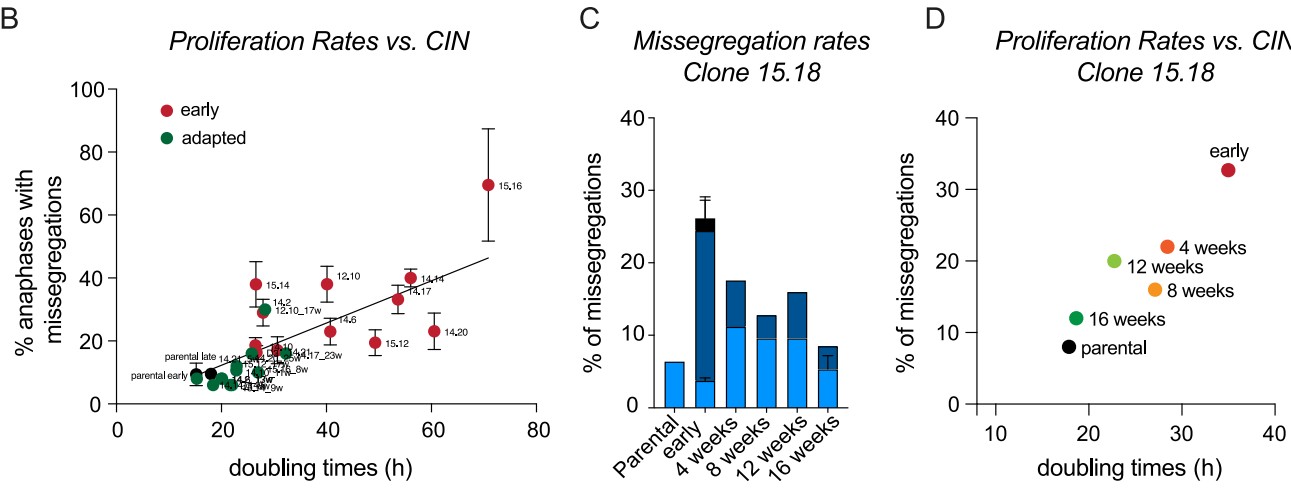

E

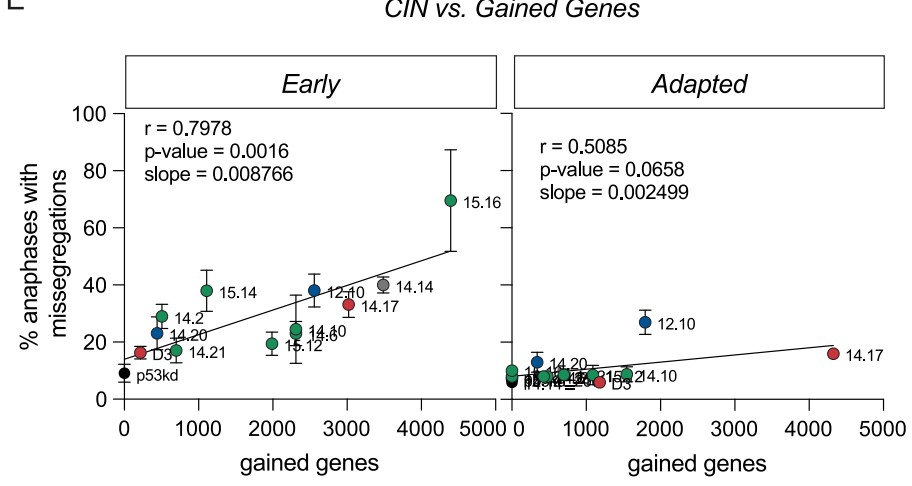

**Figure 4. Reduction of CIN is a common aspect of cells adapting to aneuploidy.**

(A) Chromosome missegregation rates determined by live-cell imaging of parental RPE-1 p53KD cells and aneuploid clones from Appendix Fig. S1A, divided into three subcategories: lagging chromosomes, anaphase bridges and others (multipolar spindle, polar chromosome, cytokinesis failure, binucleated cells). Bars are averages of at least two experiments, and a minimum of 50 anaphases were analyzed per clone per experiment. Error bars indicate standard deviation. (B) Correlation between the proliferation rates as measured in Fig. 1A and the missegregation rates as a percentage of anaphases as measured in (A). Dots represent mean, error bars indicate standard deviation. (C) Chromosome missegregation rates measured with intervals of a month for clone 15.18. Bars show mean, error bars indicate standard deviation, early $n = 2$, 16 weeks $n = 3$, in-between time points $n = 1$. (D) Correlation between proliferation rates as measured in Fig. 1A and missegregation rates as a percentage of anaphases as measured in (C) for clone 15.18. Dots represent mean, error bars indicate standard deviation. (E) Spearman correlation between the number of RNA-sequencing derived gained genes per clone and the level of CIN as determined in (A). Dots represent mean, error bars indicate standard deviation. Color-coding as determined in Fig. 2A. Source data are available online for this figure.

importantly, we found that STING can be activated in RPE-1 cells, as we observed downstream target cytokine expression using different concentrations of cGAMP (Fig. 5A). We selected the highest dose for further experiments, as all tested cytokines were upregulated, indicating the successful activation of an inflammatory response. Importantly, this activation was at least in part dependent on STING as siRNA-mediated depletion of STING suppressed the induction of cytokines induced by cGAMP (Fig. EV3B). Next, we assessed cellular fitness by measuring proliferation rates. Interestingly, we observed that parental cells treated with cGAMP have a higher doubling time (Figs. 5B and EV3C), indicating that inflammation can result in decreased proliferation. To evaluate if reducing STING-mediated inflammation in early clones could rescue the impaired proliferation phenotype, we aimed to suppress STING signaling via STING depletion. However, successful down-regulation of STING did not result in increased proliferation rates in early clones (Fig. 5C,D). Importantly, although depletion of STING reduced cytokine expression in the early clones to some extent, this reduction was clearly incomplete (Fig. EV3D), under-scoring the presence of additional inflammatory signaling pathways as shown in Fig. EV3A. In line with this, we observed that the cytokine pattern in response to cGAMP treatment is also quite distinct from the pattern we observe in the early clones. Genes that are most prominently upregulated in early clones involve INFB1, IL1B, and IL6 while upon cGAMP treatment, we find ISG15 and OAS1 to be most responsive (Fig. EV3B,EV3D). This could explain why STING depletion only marginally affects cytokine expression and further supports the notion that the CIN-induced inflamma-tion observed in early clones is multifactorial. Thus, although there is evidence for the presence of STING signaling (Fig. EV3A,D), activation of STING does not have a dominant contribution to the slow proliferation phenotype in early aneuploid clones.

## Amplification of mutant KRAS can drive adaptation

We established that reduced CIN and reduced inflammation are common aspects of adaptation to aneuploidy. We next set out to identify potential drivers of this adaptive behavior. Although we excluded karyotype simplification as a driver, karyotype alterations were commonly observed upon adaptation (Fig. 2A,B). Previous studies suggested that specific aneuploidy patterns might arise to provide survival benefits (Girish et al, 2023; Knouse et al, 2014; Mitchell et al, 2018; Taylor et al, 2018; Zack et al, 2013). In order to determine whether our clones exhibit specific aneuploidy patterns during adaptation, we examined the karyotype alterations in more detail. Strikingly, we found that 4 out of the 14 clones displayed a chromosome 12p amplification upon adaptation (Fig. 2A, clone

14.2, 14.6, 15.12 and 15.18). Moreover, two clones displayed an entire chromosome 12 amplification (clone 12.10 ad 14.20).

Importantly, one well-known oncogene, KRAS is located on 12p and one of the alleles of RPE-1 cells is mutated to an oncogenic version of KRAS (Di Nicolantonio et al, 2008). Moreover, the mutant allele of KRAS is amplified from 1 to 2 copies in parental RPE-1 cells (see micro-amplification in Appendix Fig. S1A on chr12p and (Di Nicolantonio et al, 2008)). We explored the possibility that acquiring an extra copy of the oncogenic KRAS allele provides a survival benefit to aneuploid clones. If this is true, we predict that the gained alleles consistently involve the mutant allele and not the WT. For this, we determined the relative DNA copy number of the WT and mutant alleles in parental cells as well as two of the clones displaying 12p amplification (15.12 and 15.18) and 1 clone that displayed an entire 12 amplification (12.10), using TIDE (Brinkman et al, 2014). As parental cells carry 1 WT allele and 2 mutant alleles, the expected mutant/WT ratio is 0.66. Indeed, while parental cells and early clones display the expected ratio of mutant/WT alleles, the adapted clones all display enhanced copy number of the mutant allele, suggesting that indeed the allele containing oncogenic KRAS is amplified in the adapted clones with a 12p or 12 amplification (Fig. 6A). This increased copy number also leads to enhanced expression of the mutant KRAS (Fig. 6B). To test if enhanced KRAS signaling is driving the improved proliferation in the adapted clones, we depleted KRAS using siRNA in aged parental and in adapted clones. Interestingly, although we obtained similar knockdown levels in parental cells as compared to the clones (Fig. EV4), the proliferation defect induced in the clones was much more prominent as compared to the proliferation defect observed in parental cells (Fig. 6C). This indicates that the amplification of KRAS in the adapted clones indeed contributes to adaptation.

To test whether KRAS overexpression is sufficient to drive adaptation, we infected parental and several early, non-adapted clones with a lentiviral plasmid encoding oncogenic KRAS (DD-HA-PAmCherry-HA-KRAS-G12D (Nan et al, 2015)). We con-firmed successful transduction by western blot (Fig. 6D). Strikingly, overexpressing oncogenic KRAS consistently improved prolifera-tion rates in all of the early non-adapted clones, but not in parental cells (Fig. 6E). Importantly, although proliferation rates signifi-cantly improved, they did not reach those of fully adapted clones. This suggests that oncogenic KRAS amplification can contribute, but cannot fully drive adaptation to aneuploidy. If oncogenic KRAS mutations contribute to aneuploidy tolerance, we would predict that KRAS mutant tumor cell lines have higher aneuploidy levels as they should be more tolerant. Indeed, we find a significant increase in aneuploidy scores of cell lines harboring such mutation (Fig. 6F).

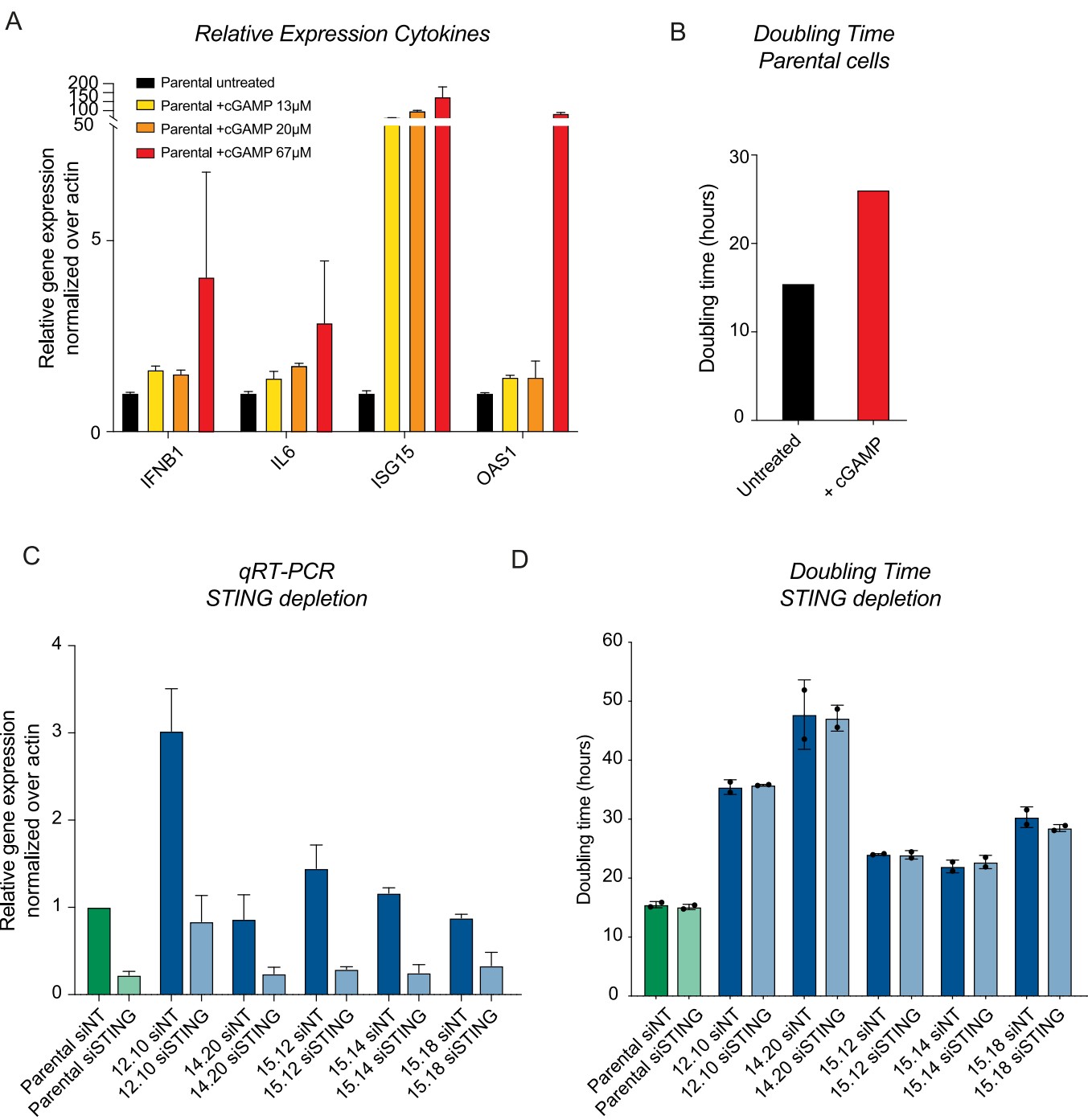

**Figure 5. Investigating the impact of STING signaling on proliferation.**

(A) mRNA levels of inflammatory response cytokines determined by qRT-PCR of parental RPE-1 p53KD, untreated or treated for 24 h with different doses of cGAMP. Values were normalized to actin and are displayed relative to expression levels in untreated parental cells. Bars show mean expression levels over three technical replicates; error bars indicate upper and lower limits. (B) Average doubling times of parental RPE-1 p53KD untreated and treated with 67 μM cGAMP added 2 h prior to imaging, determined by live-cell imaging. (C) mRNA levels of STING determined by qRT-PCR of parental RPE-1 p53KD and early aneuploid clones treated with siNT or siSTING. Values were normalized to actin and are displayed relative to expression levels in siNT parental cells. Bars show mean expression levels over three technical replicates; error bars indicate standard deviation between two experiments. (D) Doubling times of parental RPE-1 p53KD and early aneuploid clones treated with siNT or siSTING, determined via live-cell imaging. Two independent experiments were performed. Bars show mean, error bars indicate standard deviation, $n = 2$. Source data are available online for this figure.

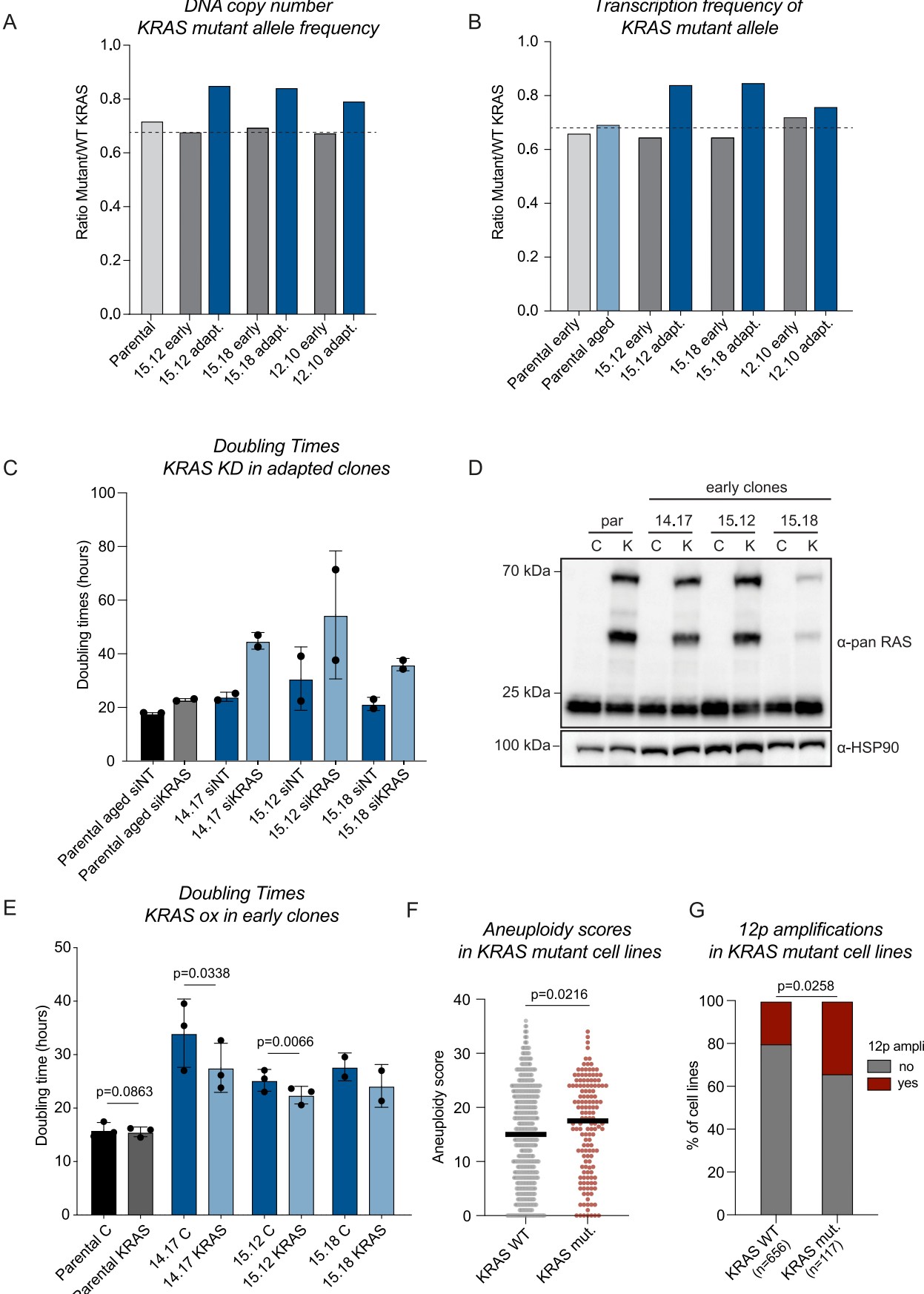

**Figure 6. Overexpression of mutant KRAS can drive adaptation.**

(A) TIDE analysis was done on isolated DNA from parental RPE-1 cells and three clones (early and adapted). As a reference DNA isolated from Hela cells was used as HeLa cells have no described KRAS mutations. The graph depicts the relative frequency of the mutant allele over the WT allele. A consistent increase in mutant allele frequency is observed upon adaptation in the three selected clones. (B) Reads extracted from the transcriptome data (GSE273576) were categorized as "mutant" or "WT" reads based on the presence of the absence of the known KRAS mutation. Only reads were quantified that span the mutant sequence and reads that could not be classified with certainty were excluded from the analysis. (C) Doubling times of parental cells and adapted clones treated with 20 nM siNT or siKRAS as determined by live-cell imaging. Bars show mean, error bars indicate standard deviation, $n = 2$. (D) Western blot analysis shows successful overexpression of KRAS in the selected early clones and parental RPE-1 cells (C= control, K= oncogenic KRAS OE). HSP90 is used as a loading control. (E) Doubling times of parental cells and early clones with and without the expression of exogenous mutant KRAS as determined by live-cell imaging. Doubling times were determined 2 weeks after the initial infection with KRAS-infected cells once selection was completed. Bars show mean, error bars indicate standard deviation, $n = 3$ except for clone 15.18 where $n = 2$. A paired *T* test was performed. (F) Aneuploidy scores were extracted from DepMap and cell lines were categorized on the presence or absence of a KRAS hotspot mutation. KRAS WT $n = 656$, KRAS mut. $n = 117$. An unpaired, two-tailed *t* test was performed (*P* value = 0.0216). (G) The presence of a chromosome 12p amplification in KRAS WT or mutant cell lines was determined on DepMap data. A Chi-square test was performed to determine the significance of the calculated odds ratio (*P* value = 0.0258). Source data are available online for this figure.

Moreover, Mutant KRAS tumor cell lines are more likely to amplify chromosome 12p, suggesting that enhancing the dose of oncogenic KRAS mutations is a beneficial trait (Fig. 6G, OR = 2.043, $P = 0.0009$).

In summary, our study has demonstrated that human cells possess the capacity to adapt to aneuploidy over time. We show that common aspects of adaptation are the reduction of CIN and the suppression of downstream concurrent inflammation and that one of the drivers of adaptation is the overexpression of oncogenic KRAS.

## Discussion

Aneuploidy is a detrimental condition for untransformed cells, but is notably prevalent in cancer cells and it is well-established that aneuploidy constitutes an early event in tumorigenesis (Bao et al, 2023; Gao et al, 2016; Gerstung et al, 2020; Krill-Burger et al, 2012; Mitchell et al, 2018; Ross-Innes et al, 2015; Teixeira et al, 2019; Wang et al, 2014). This suggests that cancer cells employ mechanisms to surmount the stress responses associated with de novo aneuploidies, enabling rapid proliferation despite aberrant karyotypes. Here, we aimed to characterize the main cellular changes during adaptation to aneuploidy induction and uncover the mechanisms underlying this adaptation in human cells in vitro. To study adaptation to aneuploidy,we selected non-transformed RPE-1 cells which are known to harbor an oncogenic KRAS mutation and a heterozygous missense mutation in CDKN2A (Di Nicolantonio et al, 2008; Libouban et al, 2017). Given the established role of p53 in aneuploidy tolerance, we used RPE-1 cells with a stable knockdown of p53. It was shown that the loss of p53 in RPE-1 cells drives anchorage-independent proliferation, and dampening of the p53-dependent transcriptional response by HRAS$^{G12V}$ in RPE-1 cells was shown to be sufficient to induce tumors in mice, indicating early tumorigenic traits (Chunduri et al, 2021). Thus, RPE-1 p53kd cells cannot be considered non-transformed but they display traits of early transformation. Furthermore, unlike cell lines isolated from advanced tumors, RPE-1 cells have not undergone full malignant transformation in vivo and have not been exposed to complex aneuploidies before. In vivo transformation typically involves selection, clonal evolution, and adaptation processes, making such cells impractical to model the adaptation response to early aneuploidy induction. Nevertheless, it is important to note that we employed an in vitro system

and future work should establish whether the findings described in this study are also relevant during early tumorigenesis in vivo.

In accordance with prior findings (Ariyoshi et al, 2016; Bonney et al, 2015; Santaguida and Amon, 2015b; Sheltzer et al, 2017; Stingele et al, 2012; Torres et al, 2007; Williams et al, 2008), the aneuploid clones generated here initially exhibited reduced proliferation rates. Strikingly, over time all of the clones displayed accelerated proliferation, often reaching or approaching parental levels, indicative of adaptation to aneuploidy. We found minor karyotype alterations in most clones, mostly towards karyotype simplification. This suggests that a selection for less complex karyotypes can contribute to the process of adaptation. However, it needs to be noted that the simplification was rather modest and a subset of clones even displayed evolved karyotypes resulting in more gene imbalances, while they still displayed decreased doubling times. Intriguingly, in adapted clones, the impact of the number of imbalanced genes on proliferation rates was notably reduced. This implies the involvement of active adaptation mechanisms facilitating better tolerance to gene imbalances.

We observed that adaptation is largely characterized by a correction of the early transcriptomic responses to aneuploidy, suggesting that the early stress responses are limiting proliferation in response to aneuploidy. Most strikingly, we observed an upregulated inflammatory response in all early clones which was consistently reduced upon adaptation. We show that the reduction of inflammation goes parallel with a reduction in CIN. Intriguingly, a recent paper identified mutations that resulted in the suppression of chromosome missegregations as a key step in adaptation to CIN in yeast (Clarke et al, 2023). This finding supports the notion that CIN exerts a toxic effect on cells and its correction is a favorable way to facilitate fast proliferation. Furthermore, there is mounting evidence that reducing CIN is a key aspect throughout tumorigenesis, even in late-stage cancers. Several studies utilizing sequencing methods to understand the evolution of tumors have reported the occurrence of "punctuated evolution", where episodes of CIN result in increased heterogeneity, followed by the selection of advantageous karyotypes, ultimately establishing clonal populations harboring low CIN (reviewed in (Bakhoum and Landau, 2017; Yates and Campbell, 2012)).

We speculate that the inflammation as a consequence of CIN contributes to its toxicity. To support this, we showed that the transcriptomic changes in early aneuploid clones, including the inflammatory responses, reflect those of parental cells experiencing acute CIN induced by mitotic checkpoint inhibition ((Hintzen et al,

2022) and Fig. EV2A). We show that inflammation as a consequence of acute CIN is complex which can be explained by the fact that CIN-induced inflammation can be prompted by multiple effects. For example, directly via micronuclear rupture or the formation of chromatin bridges resulting in cGAS/STING signaling (Dou et al, 2017; Flynn et al, 2021; MacKenzie et al, 2017; Sun et al, 2013), but also more indirectly through NF-κB and expression of SASP genes upon cellular arrest (Santaguida et al, 2017; Wang et al, 2021). STING depletion was not sufficient to overcome the proliferation defect of early aneuploid clones. It is possible that we did not suppress STING signaling sufficiently or that the duration of the knockdown was insufficient to see an effect. However, it is likely that other aspects of inflammation might be more dominant in driving reduced proliferation. Interestingly, a recent study showed that micronuclei in fact do not induce an inflammatory response via the cGAS-STING pathway but via alternative pathways such as the NF-κB pathway (Takaki et al, 2024). This is consistent with the fact that the aneuploid clones display inflammation in response to CIN while cGAS is hardly expressed. Moreover, this could explain why the reduction of STING signaling in our clones did not impact proliferation. In line with this, we find that the pattern of cytokine expression in early clones vastly differs to that of parental cells treated with cGAMP (Fig. EV2B,EV2D), and that the cytokine expression in the clones is only partially dependent on STING signaling (Fig. EV2D). Thus, it seems likely that the CIN-induced inflammation in early clones is multifactorial and due to the complex nature of this CIN-induced inflammation, it will be challenging to fully alleviate it. However, it would be valuable to fully uncover if and which aspects of the inflammatory response contribute to the impaired proliferation phenotype observed in the early aneuploid clones. In addition, while correcting CIN in early clones could provide a method to reduce inflammation, this is technically challenging, and previous attempts to do so have been unsuccessful (Hintzen et al, 2022). Nevertheless, we provide evidence that both CIN and inflammation are consistently reduced upon adaptation, and that these features are strongly linked to each other.

To understand the mechanism through which cells reduce CIN rates upon adaptation, it is important to gain more insights into the initial responses to aneuploidy that might be responsible for inducing CIN. Previously, we linked the initiation of CIN to proteotoxic stress arising as a consequence of chromosomal gains in early aneuploid clones (Hintzen et al, 2022). Consistently, in our current study, we observed indications of proteotoxic stress in early clones, which were alleviated upon adaption. These alterations consisted of a enhanced PERK-signaling as well as a reduction in processes related to translation, peptide biosynthesis, and RNA processing, responses that have been reported in other aneuploid models as well (Dürrbaum et al, 2014; Sheltzer et al, 2012; Stingele et al, 2012). Earlier studies have shown that alleviating proteotoxic stress can facilitate increased proliferation, via deletion of the deubiquitinase Ubp6 in yeast (Dephoure et al, 2014; Oromendia et al, 2012; Torres et al, 2010), or via overexpression of HSF1 in human cells (Donnelly et al, 2014), suggesting that the resolution of proteotoxic stress is crucial for adaptation. In addition to the alleviation of the UPR, we observed glycolysis and peroxisomes to be actively upregulated upon adaptation in our clones. Importantly, aneuploid cells have been shown to exhibit heightened glucose uptake (Torres et al, 2007) and aneuploidy-tolerating mutations

have been linked to metabolic rewiring. Thus, we speculate that reduced proteotoxic stress during adaptation may result from enhanced energy production via metabolic rewiring. If metabolic rewiring is indeed in place to re-establish proteosasis and if this is required for the reduction of CIN requires further investigation (Hintzen, 2024).

When evaluating large karyotype alterations, we observed a specific karyotype evolution pattern in a subset of our clones upon adaptation. This pattern concerned the gain of 12p or the entire chromosome 12 encoding oncogenic KRAS. Many studies show that specific aneuploidies can be beneficial under conditions of stress (Lukow et al, 2021; Girish et al, 2023; Ippolito et al, 2021; Knouse et al, 2017; Mitchell et al, 2018) and the observed 12p amplification is an example of a recurrent pattern that emerges in response to aneuploidy-induced stress. Our findings show that adapted clones are dependent on KRAS for proliferation and that increased expression of mutant KRAS is sufficient to drive adaptation to aneuploidy in early clones. A similar gain of chromosome 12 containing mutant KRAS was also described to occur in xenograft-derived aneuploid HCT116 cells (Girish et al, 2023), indicating that this is an event not exclusively observed in RPE-1 cells. Given the established role of KRAS in metabolic rewiring (Kerk et al, 2021), we speculate that increasing the dose of oncogenic KRAS in our cells promotes adaptation to aneuploidy by providing increased energy to potentially overcome proteotoxic stress. Interestingly, a recent study showed enhanced sensitivity of aneuploid RPE-1 clones to inhibitors of the RAF/MEK/ERK pathway (Zerbib et al, 2023). It would be relevant to test if metabolic alterations play a role in this enhanced dependency.

Collectively, our data support the following model of how cells adapt to aneuploidy in vitro (depicted in Fig. 7): as a consequence of the cellular stresses following aneuploidy induction, early aneuploid clones are highly CIN resulting in an inflammatory responses and reduced proliferation. Over time, cells adapt to aneuploidy, consistently displaying reduced CIN levels and reduced inflammation, which ultimately promotes increased proliferation rates. Interestingly a recent study also focusing on cellular adaptations to aneuploidy showed similar results in HCT116, suggesting that our findings may be applicable other cellular contexts (Boekenkamp et al, 2024). We speculate that the adaptation and reduction in CIN are a result of decreased stress responses, potentially driven by metabolic rewiring, for example, facilitated by oncogenic KRAS amplification. To gain further insights, comprehensive analyses such as full genome sequencing and proteome analysis of the adapted clones could unveil additional pathways required for adaptation. Future studies utilizing in vivo models are necessary to validate our findings and provide a more accurate understanding of cellular adaptations to aneuploidy in the context of cancer.

# Methods

## Cell culture, cell lines, and reagents

hTert-immortalized retinal pigment epithelium (RPE-1) p53kd cells were kindly provided by R. Beijersbergen. RPE-1 p53kd cells were generated by transduction with pRetroSuper-p53 (with the shRNA sequence 5′-CTACATGTGTAACAGTTCC-3′) and

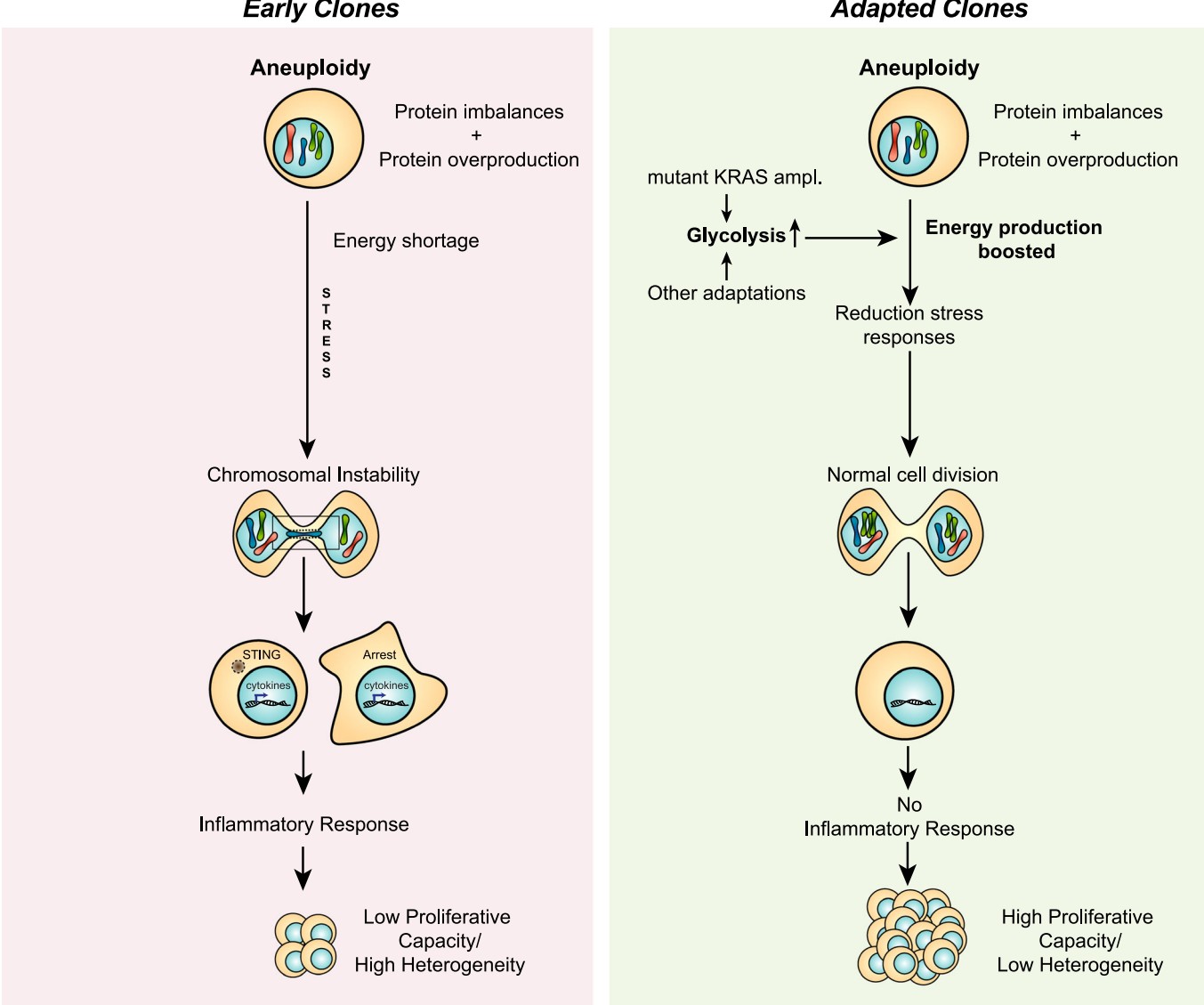

**Figure 7. Model.**

Early aneuploid cells have problems coping with the extra demands on the protein turnover machineries for the production, folding and degradation of the extra proteins encoded on the aneuploid chromosomes. This results in an excess of unfolded proteins, proteotoxic stress response and potentially additional stress responses. The aneuploidy-associated stresses lead to enhanced chromosome segregation errors. These segregation errors lead to inflammatory signaling via the STING pathway and via additional inflammatory pathways like SASP signaling. These phenotypes contribute to the slow proliferation associated to aneuploidy. Upon adaptation, aneuploidy-induced stresses are reduced, likely due to metabolic rewiring such as enhanced glycolysis. Oncogenic KRAS amplifications could contribute to metabolic rewiring but other mechanisms can drive such a switch as well. The restored energy production and reduced stress signaling prevent chromosome missegregations and inflammation and are therefore compatible with high proliferative capacity.

selected with Nutlin-3a for functional loss of p53. H2B-Dendra2 cells were made as described in (Soto et al, 2017). Cells were cultured at 37 °C at 5% $CO_2$ in Advanced Dulbecco's Modified Eagle Medium: Nutrient mixture F-12 (DMEM-F-12) with Glutamax (GIBCO), supplemented with 12% FCS (Clontech), 100 U/ml penicillin (Invitrogen), 100 μg/ml streptomycin (Invitrogen) and 2 mM UltraGlutamin (Lonza). Cells were tested for mycoplasma every 8 weeks. Inhibitors were all dissolved in DMSO and were used at the following concentrations: Mps1 inhibitor (NMS-P715) (480 μM) and a CENP-E inhibitor (GSK923295) (50 nM),

PERK inhibitor (GSK2606414) (1 μM). cGAMP was used at a concentration of 13, 20, and 67 μM.

## Generating aneuploid clones

Clones were generated by blocking RPE-1 p53kd cells in Thymidine for 14 h, after which they were released in medium containing a combination of an Mps1 inhibitor (NMS-P715) (480 μM) and a CENP-E inhibitor (GSK923295) (50 nM) for 8 h to induce whole chromosome aneuploidies and segmental aneuploidies a shown in

(Soto et al, 2017). After treatment, cells were collected by trypsinization and cells were plated single-cell in 384-well plates. On the same day, wells were examined for the presence of individual cells to ensure a single cell was present. With this approach, we successfully generated 14 aneuploid clones (Appendix Fig. S1A). Some of these clones were described in a previous study (Hintzen et al, 2022). The gain of 10q in parental RPE-1 cells, deriving from an imbalanced fusion of the q-arm of chromosomes 10 to the X chromosome ((Janssen et al, 2011), ATCC) was not considered as a de novo aneuploidy (Appendix Fig. S1A).

## Live-cell imaging

For live-cell imaging, cells were grown in a Lab-Tek II chambered coverglass (Thermo Science). Images were acquired every 5 min using a DeltaVision Elite (Applied Precision) microscope maintained at 37 °C, 40% humidity, and 5% $CO_2$, using a 20 × 0.75 NA lens (Olympus) and a Coolsnap HQ2 camera (Photometrics) with 2 times binning. DNA was visualized using SiR-DNA (Spirochrome, 0.25 μM) or using SPY-650 (Spirochrome, diluted according to the manufacturer, used 1:5000). Image analysis was done using ImageJ software, and all conditions were blinded before analysis.

## Cell proliferation analysis

Proliferation was measured by using a Lionheart FX automated microscope (Biotek). For these experiments, 500 cells were plated in 96-well plates. Two or three replicate wells were imaged per clone with a 4 h interval for 5 days, and cells were stained with the DNA dye SiR-DNA (Spirochrome, 0.25 μM) or using SPY-650 (Spirochrome, diluted according to manufacturer, used 1:5000). Proliferation rates were measured by performing cell count analysis using Gen5 software (BioTek) and doubling times were calculated using GraphPad Prism 8 software during exponential proliferation phase.

## Immunoblotting

RPE-1 cells were harvested and lysed using Laemmli buffer (120 mM Tris, pH 6.8, 4% SDS, and 20% glycerol). Equal amounts of protein were separated on a polyacrylamide gel and subsequently transferred to nitrocellulose membranes. Membranes were probed with the following primary antibodies (1:1000): pan-RAS (Rabbit, Cell Signaling, #3965), HSP90 α/β (rabbit, Santa Cruz, sc-13119). HRP-coupled secondary antibodies (Dako) were used in a 1:1000 dilution. The immunopositive bands were visualized using ECL Western blotting reagent (GE Healthcare) and a ChemiDoc MP System (Biorad).

## Copy number analysis

DNA was isolated using the DNeasy Blood and Tissue kit (Qiagen) according to the manufacturer's protocol. The number of double-stranded DNA in the genomic DNA samples was quantified by using the Qubit dsDNA HS Assay Kit (Invitrogen, cat no Q32851). Up to 2000 ng of double-stranded genomic DNA was fragmented by Covaris shearing to obtain fragment sizes of 160–180 bp. Samples were purified using 1.8× Agencourt AMPure XP PCR Purification beads according to the manufacturer's instructions (Beckman Coulter, cat no A63881). The sheared DNA samples were quantified and qualified on a BioAnalyzer system using the DNA7500 assay kit (Agilent Technologies cat no. 5067-1506). With an input of maximum, 1 μg sheared DNA, library preparation for Illumina sequencing was performed using the KAPA HTP Library Preparation Kit (KAPA Biosystems, KK8234). During library enrichment, 4–6 PCR cycles were used to obtain enough yield for sequencing. After library preparation, the libraries were cleaned up using 1× AMPure XP beads. All DNA libraries were analyzed on a BioAnalyzer system using the DNA7500 chips to determine the molarity. Up to eleven uniquely indexed samples were mixed by equimolar pooling, in a final concentration of 10 nM, and subjected to sequencing on an Illumina HiSeq2500 machine in one lane of a single read 65 bp run, according to the manufacturer's instructions. Reads have been submitted to NCBI's Sequence Read Archive (SRA) under number SRP511546.

Low-coverage whole-genome samples, sequenced single-end 65 base pairs on the HiSeq2500 were aligned to GRCh38 with bwa version 0.7, mem algorithm (Li, 2013). The mappability per 15 kilobases on the genome, for a sample's reads, phred quality 37 and higher, was rated against a similarly obtained mappability for all known and tiled 65 bp subsections of GRCh38; a reference genome-based mappability provided by QDNAseq (Scheinin et al, 2014), using a GRCh38 lifted version (https://github.com/asntech/QDNAseq.hg38.git). QDNAseq segments data using an algorithm by DNAcopy (Seshan and Olshen, 2021) and calls copy number aberrations using CGHcall (van de Wiel and Vosse, 2021), and visualization was adapted from the QDNAseq code.

## TIDE

TIDE (tracking of indels by decomposition) was used to estimate the ratio of WT KRAS to mutant KRAS shown in Fig. 6A. In short, the region harboring the 6 base pair insertion was amplified via PCR following primers: forward primer: CTTAAGCGTCGATG GAGGAG and reverse primer: TGTATCAAAGAATGGTCCTG CAC. The PCR products were treated with exonuclease I (Biolabs, M0293S) and were subjected to Sanger Sequencing using the forward primer and analyzed by the TIDE method (Brinkman et al, 2014), using WT HeLA cells as a reference, as these cells harbor WT KRAS.

## Determining the number of imbalanced genes

For clones harboring large chromosome aneuploidies, the number of imbalanced/gained/lost coding genes was calculated by first determining gained and lost segments per clone using RNA-sequencing data. The transcriptome data used for this has been deposited in NCBI's Gene Expression Omnibus under number GSE273576. A chromosomal segment was considered a gain if the average expression of genes on that segment passes a threshold of 0.31 log2 fold change (corresponding to a 1.25-fold overexpression). A segment was considered a loss if the average expression of genes on that segment was below a threshold of −0.415 log2 fold change (corresponding to 0.75-fold expression). Then, per clone the number of coding genes located on the gained or lost segments were summed up to obtain the number of lost coding or gained genes, respectively. The lost and gained coding genes were added up to obtain the total number of imbalanced genes per clone.

## RNA sequencing and data analysis

RPE-1 p53KD cells (parental and clones) were harvested in buffer RLT (Qiagen). Strand-specific libraries were generated using the TruSeq PolyA Stranded mRNA sample preparation kit (Illumina). In brief, polyadenylated RNA was purified using oligo-dT beads. Following purification, the RNA was fragmented, random-primed, and reverse transcribed using SuperScriptII Reverse Transcriptase (Invitrogen). The generated cDNA was 3′ end-adenylated and ligated to Illumina Paired-end sequencing adapters and amplified by PCR using HiSeq SR Cluster Kit v4 cBot (Illumina). Libraries were analyzed on a 2100 Bioanalyzer (Agilent) and subsequently sequenced on a Nova Seq 6000 (Illumina). We performed RNA-seq alignment using TopHat 2.1.1. on GRCh38 and counted reads using Rsubread 2.4.3 (Ensembl 102). The transcriptome data has been deposited in NCBI's Gene Expression Omnibus under number GSE273576.

We calculated differential expression between two biological replicates of the parental and each clone, as well as the Mps1 and CENP-E inhibitor treated vs control, using DESeq2 1.31.3. Copy numbers from RNA-seq data were determined using a Generalized Additive Model (GAM) smoothing (mgcv R package) with a gamma parameter of 2 and a weight parameter of 1/lfcSE. We tested for gene set differences by using a linear regression model of the Wald statistic (as reported by DESeq2) between genes belonging to a set vs. genes not belonging to a set. Gene set collections included MSigDB hallmarks (2020) and Gene Ontology (2021). For the gene sets related to inflammation, we used manually generated gene lists based on published literature. For STING signaling we used gene lists generated by Santaguida et al, 2017 (Santaguida et al, 2017), as well as a list of genes that were shown to be at least 2.5-fold differentially expressed with a $P$ value < 0.05 upon treatment with cGAMP in primary human cells (Abraham et al, 2020). For NF-κB signaling, we took a list of known NF-κB-target genes published by Boston University (https://www.bu.edu/nf-kb/gene-resources/target-genes/). For the SASP signaling, we used a recently published signature (Saul et al, 2022). See Dataset EV1 for the different gene sets.

## RNA isolation and qRT-PCR analysis

Total RNA was extracted from untreated RPE-1 cells. RNA isolation was performed by using the Qiagen RNeasy kit and quantified using NanoDrop (Thermo Fisher Scientific). cDNA was synthesized using SuperScript III reverse transcription, oligo dT (Promega), and 1000 ng of total RNA according to the manufacturer's protocol. Primers were designed with a melting temperature close to 60 degrees to generate 90–120-bp amplicons, mostly spanning introns. cDNA was amplified for 40 cycles on a cycler (model CFX96; Bio-Rad Laboratories) using SYBR Green PCR Master Mix (Applied Biosystems). Target cDNA levels were analyzed by the comparative cycle (Ct) method, and values were normalized against β-actin expression levels.

| Primer | Forward | Reverse |
|---|---|---|
| Actin | GCCGATCCACACGG AGTACTT | TTGCCGACAGGATGCA GAA |
| IFNB1 | TGCACGCTCCGGC ACTCACA | CATGGAGAACACCACTT GTTGCTC |

| Primer | Forward | Reverse |
|---|---|---|
| IL6 | GCTAGGGGTGTTTA TTGCAT | AGTGAGGAACAAGCC AGAGC |
| ISG15 | CGCAGATCACCCAGA AGATCG | TTCGCTGCATTTGTC CACCA |
| OAS1 | AGTTGACGTGCGGCT ATAAAC | GTGCTTGACTAGGCGG ATGAG |
| STING | AGACCGGTGACC ATGCTG | CGTACTCCAGGACAC AGGTG |
| KRAS | AGTGCCTTGACGA TACAGCT | CCTCCCCAGTCCTCA TGTAC |
| ATF3 | CAGTCACTGTCAGCGA CAGACCC | TCTTCTTCAGGGGCTACCTCGG |

## siRNA and crRNA transfections

siRNA transfections were performed using RNAiMAX (Invitrogen) according to the manufacturer's guidelines. The following siRNAs were used in this study: siNT (non-targeting; Dharmacon), siT-MEM173 (Dharmacon, OTP-SMARTpool), siKRAS (Dharmacon, OTP-SMARTpool). SiRNAs were used at a final concentration of 20 nM.

## Data availability

The datasets produced in this study are available in the following databases: RNA-Seq data: Gene Expression Omnibus GSE273576. Copy Number Variation (CNV) DNA Sequencing data: Sequence Read Archive (SRA): SRP511546.

The source data of this paper are collected in the following database record: biostudies:S-SCDT-10_1038-S44319-024-00252-0.

## Peer review information

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

## Acknowledgements

The authors thank the Genomics Core Facility of the Netherlands Cancer Institute for sample preparation, data acquisition, and analysis of DNA and RNA-sequencing experiments. This work was funded by a KWF (Dutch Cancer Society) Young Investigator Grant—12233 to JA Raaijmakers.

## Author contributions

**Dorine C Hintzen**: Data curation; Formal analysis; Investigation; Visualization; Writing—original draft; Writing—review and editing. **Michael Schubert**: Data curation; Formal analysis; Investigation. **Mar Soto**: Conceptualization; Data curation; Formal analysis; Investigation. **René H Medema**: Conceptualization; Supervision; Writing—review and editing. **Jonne A Raaijmakers**: Conceptualization; Supervision; Funding acquisition; Project administration; Writing—review and editing.

Source data underlying figure panels in this paper may have individual authorship assigned. Where available, figure panel/source data authorship is listed in the following database record: biostudies:S-SCDT-10_1038-S44319-024-00252-0.

## Disclosure and competing interests statement

The authors declare no competing interests.

# Expanded View Figures

**Figure EV1.  CIN is corrected during the trajectory of adaptation.**

(A) Chromosome missegregation rates determined at monthly interval by live-cell imaging of parental RPE-1 p53KD cells and aneuploid clones at from Appendix Fig. S1A, divided into three subcategories: lagging chromosomes, anaphase bridges and others (multipolar spindle, polar chromosome, cytokinesis failure, binucleated cell). All conditions were analyzed blinded. Bars are averages of at least 2 experiments and a minimum of 50 cells were filmed per clone per experiment. Error bars indicate standard deviation. (B) Spearman correlation between the number of RNA-sequencing derived imbalanced and lost genes per clone and the level of CIN as percentage of total number of anaphases as determined in Fig. 4A. Dots represent mean, error bars indicate standard deviation, $n = 1$ or 2. Color-coding as determined in Fig. 2A.

▶

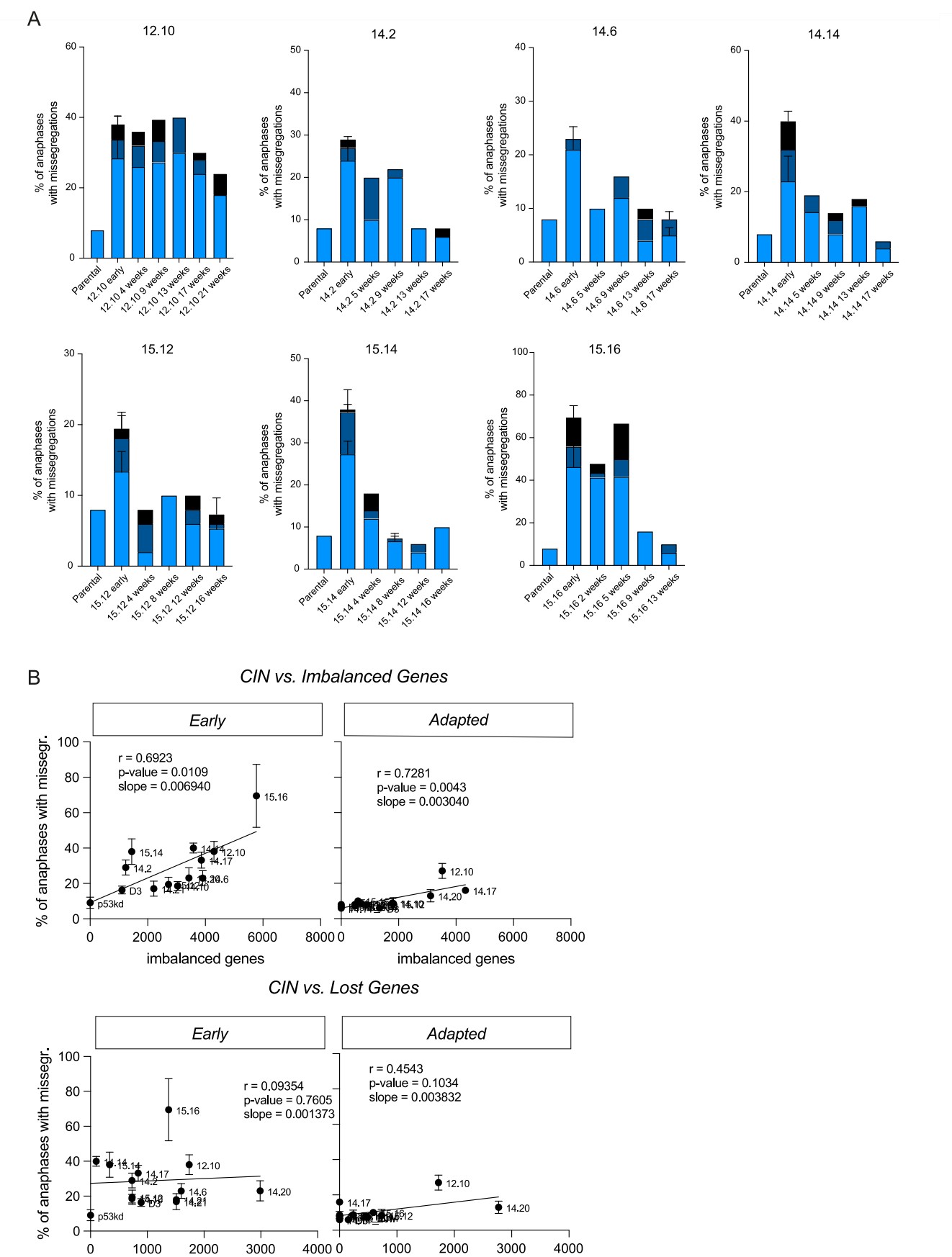

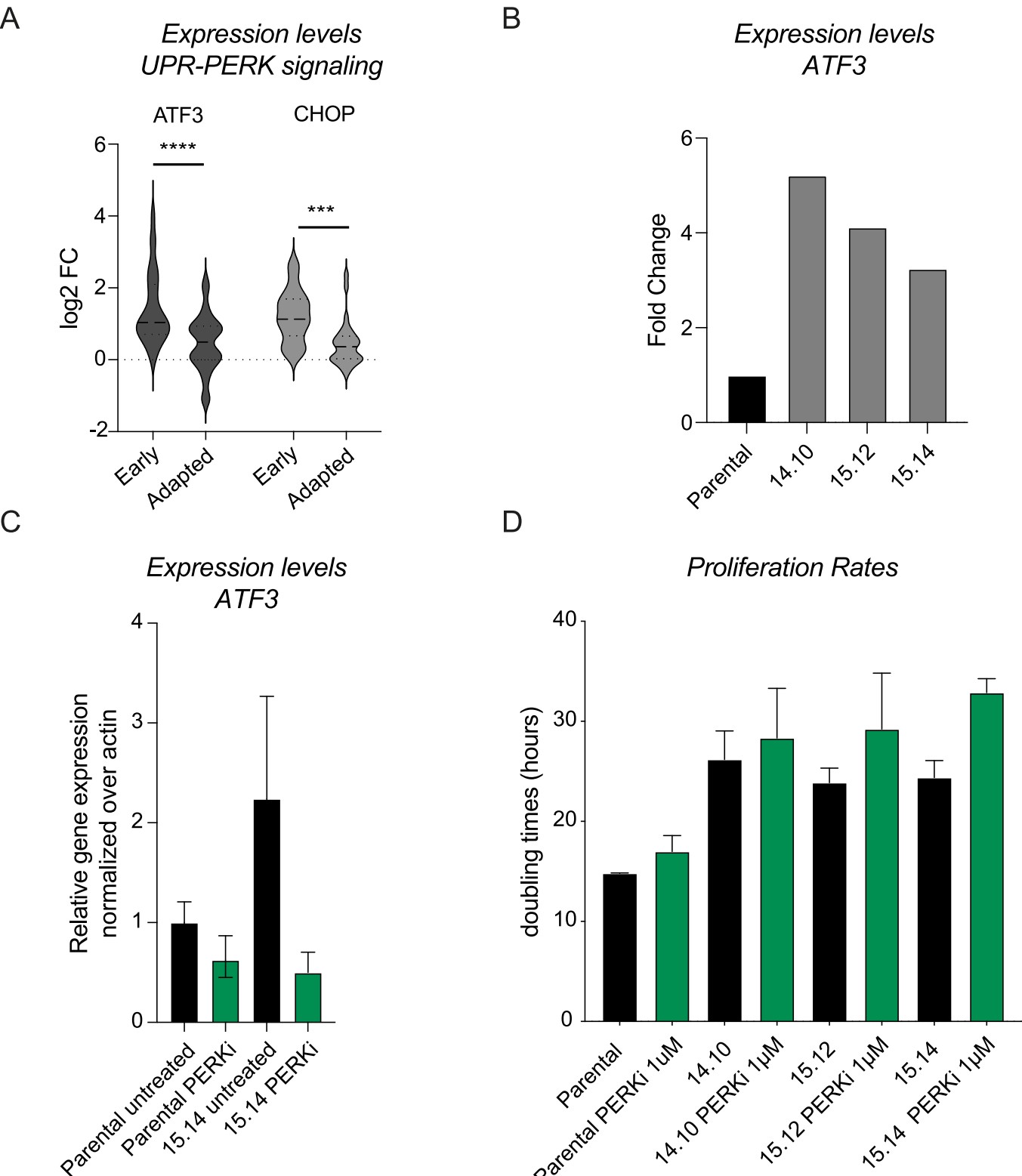

◀ **Figure EV2.  Reduced PERK-signaling is not sufficient to drive adaptation.**

(**A**) Violin plots showing the expression levels of ATF3 and CHOP in early and adapted clones extracted from the transcriptome data. An ordinary two-way ANOVA was performed between early and adapted clones ($n = 28$). $P$ values are assigned according to GraphPad standard, $P$ value for ATF3 = 0.000000002575531, $P$ value for CHOP = 0.000001051675832. (**B**) Expression level of ATF3 of early clones normalized to parental cells extracted from transcriptome data for 3 individual clones. (**C**) mRNA levels of ATF3 determined via qRT-PCR of parental RPE-1 p53KD and clone 15.14 after 24 h of treatment with 1 μM of PERK inhibitor. Values were normalized to actin and are displayed relative to expression levels in untreated parental cells. Bars show mean expression levels of 3 technical replicates; error bars indicate upper and lower limits. (**D**) Average doubling times parental RPE-1 p53KD and selected early clones untreated and treated with 1 μM PERK inhibitor added 2 h prior to imaging, determined by live-cell imaging. Error bars indicate standard deviation, $n = 2$.

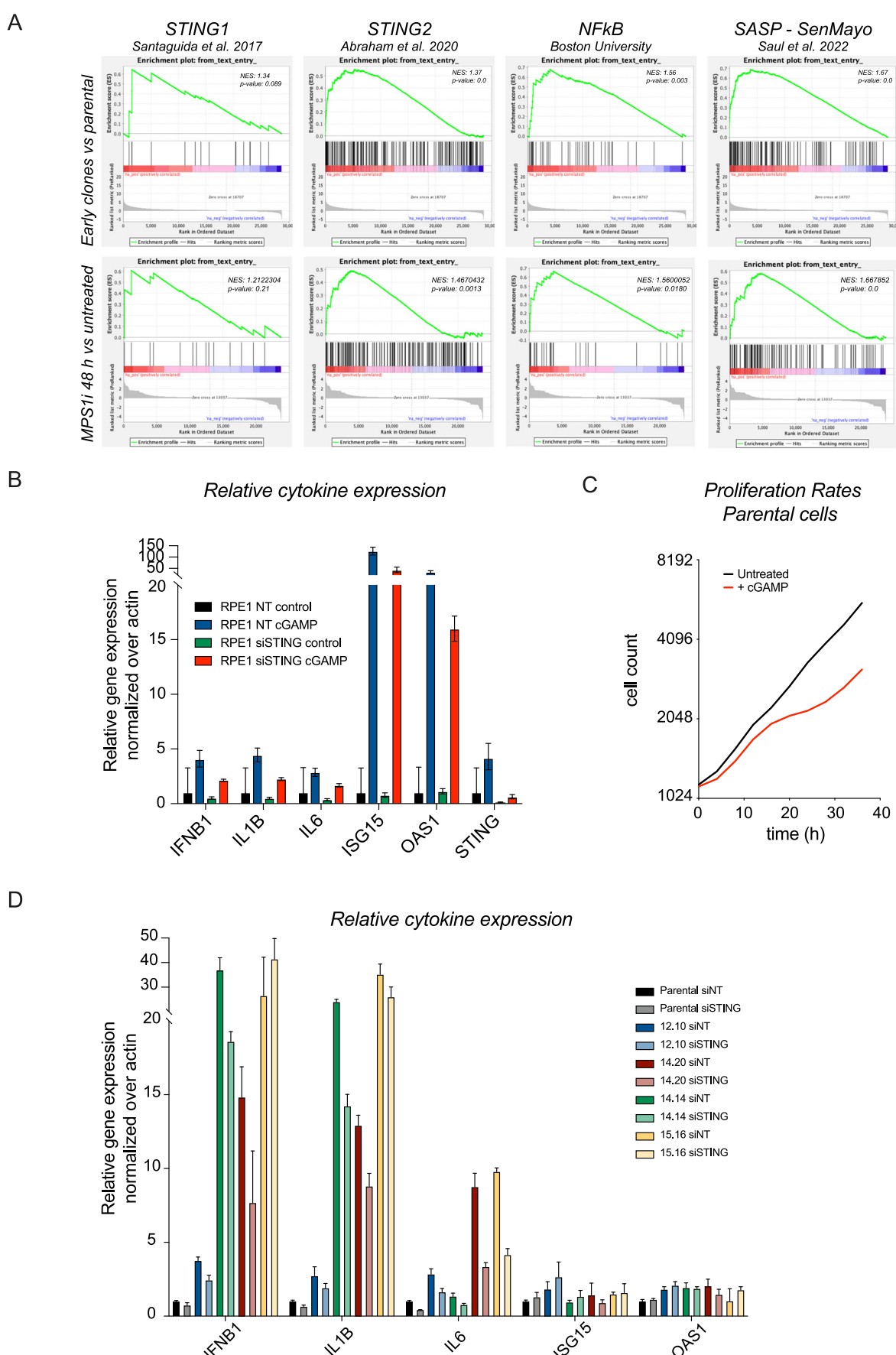

**Figure EV3. Characterization of the inflammatory response.**

(A) Transcriptome data of early clones (top row) and of parental p53KD cells treated for 48 h with 50 nM CENP-Ei and 480 nM MPS1i tested against known STING signaling, NF-κB signaling and SASP signaling gene sets. An overview of the different gene sets can be found in Dataset EV1. A weighted Kolmogorov–Smirnov statistical test was used to determine enrichment scores. (B) mRNA levels of inflammatory response cytokines determined via qRT-PCR of parental RPE-1 p53KD after 72 h siRNA against STING, untreated or treated for 24 h with 67 μM of cGAMP. Values were normalized to actin and are displayed relative to expression levels in untreated parental cells. Bars show mean expression levels of 3 technical replicates; error bars indicate upper and lower limits. (C) Cell counts of parental RPE-1 p53KD cells left untreated or treated with 67 μM cGAMP which was added 2 h prior to imaging. Y-axis Is displayed in a Log2 scale. (D) mRNA levels of inflammatory response cytokines determined via qRT-PCR. RNA of parental RPE-1 p53KD and early clones were isolated 72 h post siRNA transfection with a NT siRNA or an siRNA against STING. Values were normalized to actin and are displayed relative to expression levels in parental cells treated with non-targeting siRNA. Bars show mean expression levels; error bars indicate upper and lower limits.

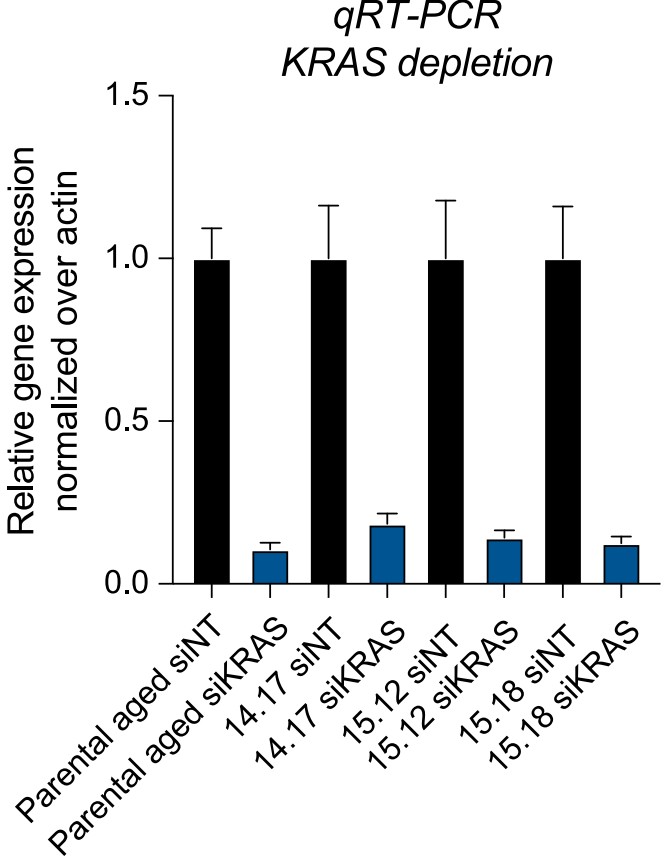

**Figure EV4. KD of KRAS in adapted clones.**

mRNA levels of KRAS determined via qRT-PCR of aged parental RPE-1 p53KD and adapted clones 24 h after siRNA transfection with a KRAS siRNA or a NT siRNA. Values were normalized to ribophorin and are displayed relative to expression levels in cells treated with siNT. Bars show mean expression levels of 3 technical replicates; error bars indicate upper and lower limits.

