## [Peer Review File · EMBO Reports]

Reduction of chromosomal instability and inflammation is a common aspect of adaptation to aneuploidy

Dorine Hintzen, Michael Schubert, Mar Soto, René Medema, and Jonne Raaijmakers

Corresponding author(s): Jonne Raaijmakers (j.raaijmakers@nki.nl) , René Medema (r.h.medema-2@prinsesmaximacentrum.nl)

Review Timeline:

Submission Date:	6th Mar 24
Editorial Decision:	28th Mar 24
Revision Received:	17th Jun 24
Editorial Decision:	12th Aug 24
Revision Received:	20th Aug 24
Accepted:	28th Aug 24

Editor: Deniz Senyilmaz Tiebe

Transaction Report: A revised version of this manuscript was transferred to EMBO reports following peer review at the EMBO Journal.

Referee #1:

In the presented manuscript, Raaijmakers and colleagues have set out to address the well-established "aneuploidy paradox" which stipulates that aneuploidy is a detrimental state for cells and organisms yet human cancers are highly aneuploid. Hintzen et al. elected to study aneuploidy tolerance in a near normal model cell line (in terms of karyotype and oncogenic potential) with suppressed p53. In their manuscript, the authors generate a number of aneuploid clones and subject them to analyses by DNA sequencing, RNA sequencing and cell biological assays. Upon induction of transient CIN and isolation of aneuploid clones that were subsequently cultured over time four classes of karyotypic changes were observed, namely evolved, simplified, reverted and more complex. The authors describe correlations between doubling times and the number of imbalanced genes. Interestingly over time in culture all clones reduced their doubling times. As expected with increased proliferation, cell cycle gene sets were differentially up-regulated in aged clones compared with young clones and inflammatory signaling was reduced. The authors attribute the changes in inflammation to a reduction in CIN as clones aged, however, a definitive link to cGAS-STING signaling could not be established, which the authors acknowledge could be a technical issue due to poor siSTING1 efficiency. Most interestingly, the authors chose to investigate the contribution of chromosome 12 gains specifically and were able to narrow it down to an increased allele frequency of the known, mutant KRAS allele of RPE-1 cells.

Overall, the study investigates an interesting phenomenon however the model system they use and the experimental setup do not adequately replicate human cancer or even in vivo contexts. This leads the authors to reach conclusions that might not reflect the reality of cancer.

Most importantly is the clear disconnect between their observation and that obtained from genomic sequencing of human tumors. Human tumors especially those that are highly aggressive and advanced can often exhibit very complex karyotypes. This suggests that reduction in CIN might not be a relevant conclusion. Specific weaknesses in the experimental system and designs include 1) Performing the work in non-transformed cells which might not have evolved to tolerate high levels of CIN 2) Doing the work on plastic which represents a very different context and environment compared to in vivo settings 3) Short duration of selection compared to in vivo context 4) Lack of therapeutic or in vivo selective pressures

Major points:

We commend the authors for their use of collected and readily available data for the assessment of karyotypes, however, we would value additional validation of the RNAseq approach by WGS that goes beyond one clone.

In Figure 3, it seems that the authors have already computed GSEA scores and since there are only four groups to compare (parental young and old, early and adapted clones), we would suggest to plot the GSEA enrichment scores instead of three of the many potential GSEA comparisons. This might also aid in understanding where the GSEA scores come from in panel B.

The link between the UPR and adaptation is weak and entirely based on the observed downregulation of two genes. Has the perturbation of either gene led to adaptation in the presence of CIN? This would make the model in Figure 7 more compelling.

The authors have gone to great lengths searching for the mechanism at play mediating the inflammatory response using siSTING1. However, these experiments are tenuous in the absence of convincing knockdown efficiency. The mRNA expression measurements are not convincing me at all since cGAMP stimulation leads to sustained cytokine induction even if slightly dampened in some cases. Many questions remain here: why was STING expression not probed bio-chemically? Why was STING not knocked out using CRISPR? Or, why did the authors not try pharmacological inhibition of STING using H-151? Furthermore, STING has been shown to be activated in a cGAS-independent manner by IFI16, thus, in the absence of cGAS in RPE-1 cells alternative mechanisms upstream of STING could be at play, too (Dunphy et al., 2018, Mol Cell). RPE1 cells are known NOT to express endogenous cGAS

The authors' observation of KRAS dosage increase is highly interesting as it provides evidence for positive selection. Indeed, the transcriptional evidence suggests a functional role of increased KRAS dosage and ectopic expression of KRAS modestly decreased doubling times. Additional validation by knocking out KRAS in adapted clones would be nice to have. Likewise, the ratio of mutant/WT KRAS suggests in 15.12 and 15.18 that up to five copies of KRAS are pre-sent per cells or that subclonal heterogeneity in KRAS copy number emerged. While formal analyses of the nature of KRAS amplification might be beyond the scope of this work their discussion might lend additional insights.

Minor point:

Have the authors considered to query CCLE to compute the gene imbalance score and perform correlative analyses with the reported doubling times to lend more cancer relevance to the story?

The visualization in Figure 2 is thoughtfully designed, but how about using the colors assigned in panel A in all subsequent figures for facile identification of the clones?

It might be worth noting that not a single clone reached the doubling time of parental cells after adaptation.

Referee #2:

This manuscript by Hintzen and colleagues, entitled "Reduction of chromosomal instability and inflammation is a common aspect of adaptation to aneuploidy", explores the mechanism by which cells adapt to CIN and aneuploidy. By measuring copy number changes, proliferation, mis-segregation rates, and gene expression over time, the authors were able to identify adaptation mechanisms. Most interesting is the decrease of inflammatory signaling over time and its

correlation with proliferation. Other interesting findings include that cells tend toward simplifying their genome after CIN and that proliferation rates correlates with number of imbalanced genes. This paper is of great interest to the EMBO readership, well-written, and experimentally sound. I only have a few points that could be worth addressing:

Major points:

- 1) I agree with the authors that there is a clear difference between slopes in Figures 2B and 2C. However, there should be a statistical test to apply to this difference, which would still be worth doing.
- 2) I am quite surprised that KRAS o/e in the parental lines (which have p53 KO/inhibited) does not increase proliferation. The authors suggest that this demonstrates a more specific effect of KRAS in the presence of CIN. To really demonstrate that, more experiments would be required including: (a) Does KRAS o/e increase glycolysis rates? (b) Does knockdown of KRAS in the cell lines with higher KRAS decrease proliferation again? (c) Does inhibition of glycolysis counteract the KRAS effect? Another possibility is that the parental cells are already at the max proliferation rate, so the KRAS o/e phenotype is not observed (rather than this being related to CIN). The authors could perform some additional experiments or soften the language around interpreting Figure 6. (The KRAS experiments are not necessary for the key points of this paper.)
- 3) All experiments here are in one system (RPE; p53^{-/-}). I do not think the authors need to perform these experiments in other cell lines. However, they could add some text in the discussion about whether they expect the same to be true in other cell contexts.

Minor points:

- 1) In the paper, the authors referred to reference 40 for how they induced aneuploidy. I think this should be spelled out in the results section as well.
- 2) There are figure typos, including (but not limited too):
 - a. Figure 5: Some panels are not referred to in the text (5D), or they are referred to out of order.
 - b. Figure 2: Wrong panels are sometimes referred to in the text.
 - c. Figure 3: In panel 3C, heading should say "GSEA" rather than "GSE".

Referee #3:

Aneuploidy is a hallmark of cancer, a disease characterised by uncontrolled proliferation, yet it is very detrimental in untransformed cells. This suggests that aneuploid cancer cells can tolerate aneuploidy-associated cellular stresses. The study by Hintzen et al. identify the mechanisms underlying this adaptation and report that reduction of chromosomal instability and inflammation is a key requirement in this process. Further they found that amplification of KRAS can promote adaptation to aneuploidy. The authors made several interesting observations, the data presented are convincing and the experiments well-done. The paper is well written, it reads well and the main conclusions are compelling and novel. Thus, the paper is suitable for publication in The EMBO Journal. This Reviewer has only minor points, listed below, that can be helpful in improving the work:

1. In the introduction, when discussing dosage compensation mechanism, it might be worth citing PMID: 35575458
2. Also, when discussing proteotoxic stress, it might be worth citing PMID: 26404941
3. When the authors describe the contribution of aneuploidy to heterogeneity, tumor progression, etc., it might be worth citing PMID: 34266888 and 34266887
4. In the first paragraph of the results, it is worth adding that RPE1 cells are untransformed
5. Also in the first paragraph of results, last sentence: "...all clones initially displayed a growth defect..." it would be more appropriate to use "proliferation defect" rather than "growth defect", unless the authors actually measured growth rather than proliferation. The same applies to the beginning of page 8 ("To evaluate if reducing cGAS/STING-mediated inflammation in early clones could rescue the slow growth phenotype").
6. It looks like reference #6 and #74 are duplicated.

Dear Jonne,
Dear René,

Thank you for transferring your manuscript along with referee reports to EMBO Reports, which was previously reviewed at another journal. I have now read your preliminary point-by-point response to referee concerns carefully. I appreciate that you can address many of the concerns raised experimentally and by performing textual changes, and see that the proposed revisions will strengthen the manuscript.

As per your point regarding the KRAS depletion experiments, I agree with you and referee #2 that the interpretations of Figure 6 needs to be toned down in case KRAS turns out to be essential for clone survival.

Regarding the UPR experiments, if the results of UPR interference turn out to be negative, please do keep the data in the manuscripts but move it to the EV Figures and include a discussion point about them in the text.

Lastly, regarding the STING aspect, I agree that given the existing knock down and inhibitor experiments, and that STING is not proposed to be the main driver of inflammation and decreased growth rate, performing additional knock out analysis is not required.

Having looked at everything, I would like to invite you to submit a revised manuscript. However, I would like to point out that we need strong support from the referees following revision to consider publication here. It is this aspect that is more difficult to assess at this stage.

Please see the guidelines for the revision below my signature.

I look forward to seeing a revised version of your manuscript when it is ready. Please let me know if you have questions or comments regarding the revision.

Kind regards,

Deniz

Deniz Senyilmaz Tiebe, PhD
Editor
EMBO Reports

--

Please revise your manuscript with the understanding that the referee concerns (as in their reports) must be fully addressed and their suggestions taken on board. Please address all referee concerns in a complete point-by-point response. Acceptance of the manuscript will depend on a positive outcome of a second round of review. It is EMBO reports policy to allow a single round of major experimental revision only and acceptance or rejection of the manuscript will therefore depend on the completeness of your responses included in the next, final version of the manuscript.

We realize that it is difficult to revise to a specific deadline. In the interest of protecting the conceptual advance provided by the work, we recommend a revision within 3 months. Please discuss the revision progress ahead of this time with me if you require more time to complete the revisions, or if you have questions or comments regarding the revision (also by video chat).

1. A data availability section providing access to data deposited in public databases is missing (where applicable).
2. Your manuscript contains statistics and error bars based on $n=2$. Please use scatter plots in these cases.

You can submit the revision either as a Scientific Report or as a Research Article. For Scientific Reports, the revised manuscript can contain up to 5 main figures and 5 Expanded View figures, and it should not exceed 27000 characters. If the revision leads to a manuscript with more than 5 main figures it will be published as a Research Article. In this case the Results and Discussion section should be separate. If a Scientific Report is submitted, these sections have to be combined. This will help to shorten the manuscript text by eliminating some redundancy that is inevitable when discussing the same experiments twice. In either case, all materials and methods should be included in the main manuscript file.

4) a .docx formatted letter INCLUDING the reviewers' reports and your detailed point-by-point responses to their comments. As part of the EMBO publication's Transparent Editorial Process, EMBO reports publishes online a Review Process File (RPF) to accompany accepted manuscripts. This File will be published in conjunction with your paper and will include the referee reports, your point-by-point response and all pertinent correspondence relating to the manuscript.

<https://www.embopress.org/page/journal/14693178/authorguide#transparentprocess>

5) a complete author checklist, which you can download from our author guidelines

<https://www.embopress.org/page/journal/14693178/authorguide>. Please insert information in the checklist that is also reflected in the manuscript. The completed author checklist will also be part of the RPF.

6) Please note that all corresponding authors are required to supply an ORCID ID for their name upon submission of a revised manuscript (<https://orcid.org/>). Please find instructions on how to link your ORCID ID to your account in our manuscript tracking system in our Author guidelines

<https://www.embopress.org/page/journal/14693178/authorguide#authorshipguidelines>

7) Before submitting your revision, primary datasets produced in this study need to be deposited in an appropriate public database (see <https://www.embopress.org/page/journal/14693178/authorguide#datadeposition>). Please remember to provide a reviewer password if the datasets are not yet public. The accession numbers and database should be listed in a formal "Data Availability" section placed after Materials & Method (see also

<https://www.embopress.org/page/journal/14693178/authorguide#datadeposition>). Please note that the Data Availability Section is restricted to new primary data that are part of this study. * Note - All links should resolve to a page where the data can be accessed. *

Additional information on source data and instruction on how to label the files are available:

<https://www.embopress.org/page/journal/14693178/authorguide#sourcedata>

9) Our journal encourages inclusion of *data citations in the reference list* to directly cite datasets that were re-used and obtained from public databases. Data citations in the article text are distinct from normal bibliographical citations and should directly link to the database records from which the data can be accessed. In the main text, data citations are formatted as follows: "Data ref: Smith et al, 2001" or "Data ref: NCBI Sequence Read Archive PRJNA342805, 2017". In the Reference list, data citations must be labeled with "[DATASET]". A data reference must provide the database name, accession number/identifiers and a resolvable link to the landing page from which the data can be accessed at the end of the reference. Further instructions are available at <http://www.embopress.org/page/journal/14693178/authorguide#referencesformat>

12) Please also note our reference format:
<http://www.embopress.org/page/journal/14693178/authorguide#referencesformat>

Point-by-point response

Referee #1:

In the presented manuscript, Raaijmakers and colleagues have set out to address the well-established "aneuploidy paradox" which stipulates that aneuploidy is a detrimental state for cells and organisms yet human cancers are highly aneuploid. Hintzen et al. elected to study aneuploidy tolerance in a near normal model cell line (in terms of karyotype and oncogenic potential) with suppressed p53. In their manuscript, the authors generate a number of aneuploid clones and subject them to analyses by DNA sequencing, RNA sequencing and cell biological assays. Upon induction of transient CIN and isolation of aneuploid clones that were subsequently cultured over time four classes of karyotypic changes were observed, namely evolved, simplified, reverted and more complex. The authors describe correlations between doubling times and the number of imbalanced genes. Interestingly over time in culture all clones reduced their doubling times. As expected with increased proliferation, cell cycle gene sets were differentially up-regulated in aged clones compared with young clones and inflammatory signaling was reduced. The authors attribute the changes in inflammation to a reduction in CIN as clones aged, however, a definitive link to cGAS-STING signaling could not be established, which the authors acknowledge could be a technical issue due to poor siSTING1 efficiency. Most interestingly, the authors chose to investigate the contribution of chromosome 12 gains specifically and were able to narrow it down to an increased allele frequency of the known, mutant KRAS allele of RPE-1 cells.

Overall, the study investigates an interesting phenomenon however the model system they use and the experimental setup do not adequately replicate human cancer or even in vivo contexts. This leads the authors to reach conclusions that might not reflect the reality of cancer.

We thank the reviewer for taking the time to review our paper and for his/her comments and suggestions.

We agree that the effects we observed do not necessarily hold true in cancer cells. In fact, we realize that we did not emphasize sufficiently that our aim was not to replicate human cancers but rather to establish a simplified system that mimics early-stage cancers or even pre-cancerous lesions, encountering an initial event of aneuploidy, something that is extremely challenging to model in an in vivo setting. Nevertheless, we agree that our in vitro system lacks direct translatability to the in vivo cancer setting. Therefore, we now clearly state this on two occasions in the discussion (first paragraph and final paragraph) and describe on several occasions that our model system reflects adaptation in vitro to not make any statement on the translatability to the in vivo setting. Below we comment more specifically on the 4 specified weaknesses in the paper and how we addressed these issues in the revised version.

Most importantly is the clear disconnect between their observation and that obtained from genomic sequencing of human tumors. Human tumors especially those that are highly aggressive and advanced can often exhibit very complex karyotypes. This suggests that reduction in CIN might not be a relevant conclusion.

The reviewer is right in that highly aggressive tumors often contain very complex karyotypes. Yet, it is interesting to note that CIN and complex karyotypes are not always inherently correlated. In fact, there is mounting evidence that reducing CIN is a key aspect throughout tumorigenesis, even in late-stage cancers. Several studies utilizing sequencing methods to map the evolution of tumors have reported the occurrence of "punctuated evolution". These episodes, also referred to as "bursts of CIN", result in increased heterogeneity, followed by the selection of advantageous karyotypes, ultimately establishing clonal populations with increasing complexity but with restricted heterogeneity (reviewed in Bakhoun and Landau 2017; Knouse et al. 2017; Yates and Campbell 2012). Therefore, we believe the argument that reduction of CIN is not a relevant conclusion in

relation to cancer is not necessarily true. We discussed the potential translation to advanced cancers better in the discussion section, of course taking in account the limitations of our model system.

Specific weakness in the experimental system and designs include 1) Performing the work in non-transformed cells which might not have evolved to tolerate high levels of CIN 2) Doing the work on plastic which represents a very different context and environment compared to in vivo settings 3) Short duration of selection compared to in vivo context 4) Lack of therapeutic or in vivo selective pressures

We understand the concerns about specific weaknesses of our experimental design, but we will address these concerns accordingly:

1) We utilized RPE-1 cells with a stable knock-down of p53. We agree that these cells unlikely tolerate high levels of CIN. However, this is in fact one of the reasons that we selected this cell type and we do not see this as a weakness but rather as a strength (also see response to point 2). Cell lines derived from advanced tumors have likely already undergone adaptation to karyotypic alterations and/or enhanced levels of CIN. This makes them therefore less suitable to model adaptive behavior. Thus, we selected a cell line that rather reflects an early transformation setting, as that is the moment during tumorigenesis when the first aneuploidies are introduced. Importantly, in addition to p53-deficiency, RPE-1 cells harbor an oncogenic KRAS mutation as well as a missense mutation in CDKN2A (Libouban et al. 2017; Di Nicolantonio et al. 2008). TP53 loss in RPE-1 cells has been demonstrated to be sufficient to drive anchorage-independent growth, indicating that these cells display characteristics that are common to transformed cells (Chunduri et al. 2021). Moreover, dampening of the p53-dependent transcriptional response by HRAS^{G12V} in RPE-1 cells is sufficient to induce tumors in mice (Segeren et al. 2022). Therefore, we conclude that these cells cannot be considered completely untransformed but rather reflect early tumorigenesis/transformation. We explained the rationale of selecting this model more clearly in the manuscript.

2) This is a valid point and there are indeed important differences between experiments performed in vitro and in vivo and therefore, the translation of our in vitro experiments indeed needs to be done carefully. There certainly are specific advantages of using our in vitro system. Unlike cancer cells, our RPE-1 cells have not undergone fully malignant transformation in vivo. In vivo transformation typically involves selection, clonal evolution, and adaptation processes, making it impractical for modeling the adaptation response to early aneuploidy induction. It is widely established that aneuploidy constitutes an early event in tumorigenesis (Bao et al. 2023; Gao et al. 2016; Gerstung et al. 2020; Krill-Burger et al. 2012; Mitchell et al. 2018; Ross-Innes et al. 2015; Teixeira et al. 2019; Wang et al. 2014). Our model system serves the purpose of comprehending the fundamental cellular processes underlying adaptation to this initial exposure to aneuploidy, without the additional complexities associated with in vivo models. This warrants the use of clean cellular systems that are traceable. Moreover, such a cell system allows for reliable quantification of traits such as CIN, which is much more complicated in tumors. Nevertheless, we agree that we need to be careful with the direct translation to an in vivo cancer setting. Consequently, we rewrote the manuscript from a more fundamental perspective and tempered certain conclusions directly tied to cancer biology.

3) We cultured our clones for over 7 months in the lab. Compared to tumor growth rates in certain mouse models or even in patients, this might be a relevant time-span for selection to occur in vivo. However, as discussed in the previous point, in vivo contexts are much more complex and clonal selection rates might harbor different kinetics in vivo as compared to in vitro and therefore our data might not be directly translatable to in vivo settings. We added this notion to the discussion of our paper.

4) The aim of our study is to uncover how cells can cope and adapt to an initial encounter with aneuploidy. Adding selective pressure will add an extra layer of complexity to this question, and is therefore more relevant for future studies.

Major points:

We commend the authors for their use of collected and readily available data for the assessment of karyotypes, however, we would value additional validation of the RNAseq approach by WGS that goes beyond one clone.

We agree with the reviewer that WGS data of more adapted clones would be valuable, and we added the data of three additional clones (both early and adapted) to the manuscript. This data showed that indeed RNAseq data can be used to reliably extract karyotype information.

In Figure 3, it seems that the authors have already computed GSEA scores and since there are only four groups to compare (parental young and old, early and adapted clones), we would suggest to plot the GSEA enrichment scores instead of three of the many potential GSEA comparisons. This might also aid in understanding where the GSEA scores come from in panel B.

We thank the reviewer for this suggestion and we included the GSEA scores to the supplemental data.

The link between the UPR and adaptation is weak and entirely based on the observed downregulation of two genes. Has the perturbation of either gene led to adaptation in the presence of CIN? This would make the model in Figure 7 more compelling.

We thank the reviewer for this suggestion. We aimed to knock-down both ATF3 as well as CHOP using siRNA in early clones and measure proliferation rates to see if adaptation occurs. Unfortunately, CHOP knockdowns were unsuccessful and we only managed to knockdown ATF3 partially in some clones, however we did not observe any prominent effects on proliferation rates (see figure below). As an alternative approach, we tried to downregulate the PERK-arm of the UPR pharmacologically by treating cells with a specific PERK-inhibitor (GSK2606414). To test the efficacy of the inhibitor we performed qRT-PCR and we showed that treatment with the PERK inhibitor efficiently repressed the enhanced ATF3 levels in an early clone (new Fig. S4C). Nevertheless, treatment with the inhibitor did not improve proliferation rates in our clones but actually often led to decreased growth rates (new Fig. S4D).

These data suggest that although PERK-mediated UPR is activated in early clones, this is not the key signaling response that is responsible for the slow proliferation. We now added this data to the supplemental figures (Fig. S4C,D) and adjusted the model and our conclusions accordingly.

The authors have gone to great lengths searching for the mechanism at play mediating the in-

inflammatory response using siSTING1. However, these experiments are tenuous in the absence of convincing knockdown efficiency. The mRNA expression measurements are not convincing to me at all since cGAMP stimulation leads to sustained cytokine induction even if slightly dampened in some cases. Many questions remain here: why was STING expression not probed bio-chemically? Why was STING not knocked out using CRISPR? Or, why did the authors not try pharmacological inhibition of STING using H-151? Furthermore, STING has been shown to be activated in a cGAS-independent manner by IFI16, thus, in the absence of cGAS in RPE-1 cells alternative mechanisms upstream of STING could be at play, too (Dunphy et al., 2018, Mol Cell). RPE1 cells are known NOT to express endogenous cGAS

Indeed, we set out to investigate the role of STING in the inflammatory response by using siRNA mediated knock-down. We used a knock-down strategy as CRISPR KO would require picking of individual clones which could induce a bias towards faster proliferating clones, bringing in complications.

However, we agree with the reviewer that knock-down of STING did not fully rescue the induced cytokine expression to baseline levels. To overcome this, we previously already tested two different pharmacological STING inhibitors: H-151 and SN-011. However, unfortunately, in our system these inhibitors did not rescue the induced cytokine expression at all and in some cases even causes inflammation (See figures below), making these inhibitors unsuitable as a strategy to reduce STING-signaling in our model system. Therefore, we also did not include this data in the manuscript.

Regarding the concerns of incomplete knockdown-efficiency using siRNA we do want to note that cytokine levels were more than slightly dampened with a 50% reduction in IFNB1, IL1B, IL6 and OAS1 and a 70% reduction in ISG15. Nevertheless, knockdown of STING was not sufficient to improve growth in the early clones.

To further investigate this, we studied the extent of cytokine induction in early clone compared to cGAMP treated cells. Importantly, we observed that in the early clones, the cytokine pattern is distinct from the pattern we observe as a consequence of cGAMP treatment. For example, genes that are most prominently upregulated in early clones involve INFB1, IL1B and IL6 while upon cGAMP treatment we find ISG15 and OAS1 to be most responsive (see new Figure S5D compared to S5B). This further supports the notion that the inflammation in early clones is unlikely driven solely

by STING. In line with this, knockdown of STING only partially prevented the high expression of IFNB1, IL1B and IL6 in the early clones. Indeed, as described in our original submission, we find evidence that our clones display signs of SAPS-signaling and NFkB-signaling (Fig. S5), pathways that are at play independent of cGAS/STING. Thus, although we cannot formerly exclude that upon complete knockdown of STING we would observe an effect on growth, we provide evidence that our clones harbor a more complex and multifactorial inflammatory response that can contribute to the slow growth phenotype. We did add the different cytokine profiles from cGAMP treated cells and early clones to the revised manuscript to further support the notion that STING signaling is not the sole cause for the observed inflammatory response. Interestingly, a very recent publication showed that micronuclei that arise as a consequence of CIN do not trigger a cGAS-STING response, but rather induce inflammation via alternative pathways (Takaki et al, 2024). This fits with our data that cytokine profiles differ between early clones and cGAMP-treated cells and would also explain why interfering with STING is not sufficient to rescue proliferation. We included this reference in our discussion section. Finally, since our data indeed suggests that RPE-1 hardly express cGAS and there are alternative routes to STING activation as also pointed out by this reviewer, we now refer to 'STING-signaling' instead of 'cGAS/STING signaling' when interfering with STING in our system.

The authors' observation of KRAS dosage increase is highly interesting as it provides evidence for positive selection. Indeed, the transcriptional evidence suggests a functional role of in-creased KRAS dosage and ectopic expression of KRAS modestly decreased doubling times. Additional validation by knocking out KRAS in adapted clones would be nice to have. Likewise, the ratio of mutant/WT KRAS suggests in 15.12 and 15.18 that up to five copies of KRAS are pre-sent per cells or that subclonal heterogeneity in KRAS copy number emerged. While formal analyses of the nature of KRAS amplification might be beyond the scope of this work their dis-cussion might lend additional insights.

We thank the reviewer for his/her suggestion and in response to this suggestion, we performed knock-down experiments using a SMARTpool siRNA against KRAS. Interestingly, we observed that knockdown of KRAS only mildly affected proliferation rates of parental RPE-1 cells while proliferation was much more severely affected in adapted clones. This further underscores the contribution of KRAS to the adaptation process. We included this piece of data to our manuscript (Fig. 6C and S6).

Regarding the ratio of mutant and WT KRAS in adapted clones, when a clone contains 4 copies of KRAS this would result in a WT/mutant ratio of 0.8. We agree with the reviewer that from the plots, clones 15.12 and 15.18 have a slightly higher ratio than 0.8; however, we cannot exclude that this is due to experimental noise. As 5 mutant copies would result in a ratio of 0.83, these experiments are not sensitive enough to distinguish between 4 or 5 copies of mutant KRAS.

Minor point:

Have the authors considered to query CCLE to compute the gene imbalance score and perform correlative analyses with the reported doubling times to lend more cancer relevance to the sto-ry?

We thank the reviewer for this interesting suggestion. We indeed considered to take the CCLE data into account, however we speculated that although a correlation could be present between gene imbalances and doubling times, this data is complex to interpret. It is very likely that cancer cell lines have undergone adaptation to karyotype aberrations to some extent, both in vivo and in vitro. Nevertheless, we think future work could be done on this. For example, stratifying the cell lines that are most deviating from the predicted growth defects based on their karyotype deviations would represent cell lines that are adapted to harboring abnormal karyotypes. Finding similarities amongst these cell lines in terms of mutations, aneuploidy patterns or transcriptome alterations could provide hints to the adaptation mechanisms in tumors. Although an extremely interesting direction,

we feel that currently this is out of the scope of the manuscript, also since we toned down our conclusions regarding the translation to cancer.

The visualization in Figure 2 is thoughtfully designed, but how about using the colors assigned in panel A in all subsequent figures for facile identification of the clones?

We thank the reviewer for this suggestion and we implemented this color-coding in our subsequent figures.

It might be worth noting that not a single clone reached the doubling time of parental cells after adaptation.

Indeed, there are some clones that do not reach the doubling time of parentals (clone 12.10, 14.16, 14.17 and 14.20). However, other clones do reach the doubling time of parentals (Figure 1B: clone 14.2, 15.12, 15.14, 15.16). Possibly, the reviewer is referring to some small remaining differences that are sometimes observed that are in the range of 1-2 hours compared to parentals. However, since replicates are not always present, statistics to proof a real difference are not possible. Therefore, we do not dare to make strong conclusions on these examples. However, as suggested by this reviewer, we mentioned the clones with a more prominent remaining growth defect in the revised paper.

Referee #2:

This manuscript by Hintzen and colleagues, entitled "Reduction of chromosomal instability and inflammation is a common aspect of adaptation to aneuploidy", explores the mechanism by which cells adapt to CIN and aneuploidy. By measuring copy number changes, proliferation, mis-segregation rates, and gene expression over time, the authors were able to identify adaptation mechanisms. Most interesting is the decrease of inflammatory signaling over time and its correlation with proliferation. Other interesting findings include that cells tend toward simplifying their genome after CIN and that proliferation rates correlates with number of imbalanced genes. This paper is of great interest to the EMBO readership, well-written, and experimentally sound. I only have a few points that could be worth addressing:

We thank the reviewer for the kind remarks on our manuscript and are pleased to see that he/she acknowledged that our findings are of broad interest.

Major points:

1) I agree with the authors that there is a clear difference between slopes in Figures 2B and 2C. However, there should be a statistical test to apply to this difference, which would still be worth doing.

We thank the reviewers for their suggestion and we performed the required statistical analysis.

2) I am quite surprised that KRAS o/e in the parental lines (which have p53 KO/inhibited) does not increase proliferation. The authors suggest that this demonstrates a more specific effect of KRAS in the presence of CIN. To really demonstrate that, more experiments would be required including: (a) Does KRAS o/e increase glycolysis rates? (b) Does knockdown of KRAS in the cell lines with higher KRAS decrease proliferation again? (c) Does inhibition of glycolysis counteract the KRAS effect? Another possibility is that the parental cells are already at the max proliferation rate, so the KRAS o/e phenotype is not observed (rather than this being related to CIN). The authors could perform some additional experiments or soften the language around interpreting Figure 6. (The KRAS experiments are not necessary for the key points of this paper.)

We thank the reviewer for their suggestions on experiments related to KRAS and agree that some additional experiment would be valuable. Most importantly, we knocked-down KRAS using siRNA in adapted clones and parental cells and evaluated the effect by measuring proliferation rates (also see response to reviewer 1 regarding this point). We found that knockdown of KRAS led to impaired proliferation in the adapted clones, underlining the role of KRAS in facilitating increased proliferation during adaptation. We included add this data to our manuscript.

Regarding the glycolysis experiments, we tried to investigate glycolysis rates in the KRAS overexpression clones but failed to obtain accurate and reproducible measurements, partly due to differences in proliferation rates between clones and partly due because we could not determine extent of inhibition of glycolysis between different measurements when using inhibitor of glycolysis. Since the link between KRAS, glycolysis and CIN was purely speculative, we decided to soften the language around these data and clarify that these are speculations that require further investigation.

3) All experiments here are in one system (RPE; p53^{-/-}). I do not think the authors need to perform these experiments in other cell lines. However, they could add some text in the discussion about whether they expect the same to be true in other cell contexts.

We kindly thank the reviewer for this suggestion and have addressed this point in our discussion. Intriguingly, a recent preprint on Biorxiv by (Boekenkamp et al. 2024) showed similar results to our results, both in RPE-1 cells as well as in HCT116 cells. As the results of the RPE-1 cells overlap largely with the HCT116, we expect our conclusions based on RPE-1 also to be valid in other cellular contexts. We elaborated on this in our discussion.

Minor points:

1) In the paper, the authors referred to reference 40 for how they induced aneuploidy. I think this should be spelled out in the results section as well.

We elaborated on the method of aneuploidy induction in our results section.

2) There are figure typos, including (but not limited too): We apologize for the mistakes and we will corrected this in our text. We thoroughly scanned the text for additional mistakes and corrected them.

a. Figure 5: Some panels are not referred to in the text (5D), or they are referred to out of order.

b. Figure 2: Wrong panels are sometimes referred to in the text.

c. Figure 3: In panel 3C, heading should say "GSEA" rather than "GSE".

Referee #3:

Aneuploidy is a hallmark of cancer, a disease characterised by uncontrolled proliferation, yet it is very detrimental in untransformed cells. This suggests that aneuploid cancer cells can tolerate

aneuploidy-associated cellular stresses. The study by Hintzen et al. identify the mechanisms underlying this adaptation and report that reduction of chromosomal instability and inflammation is a key requirement in this process. Further they found that amplification of KRAS can promote adaptation to aneuploidy. The authors made several interesting observations, the data presented are convincing and the experiments well-done. The paper is well written, it reads well and the main conclusions are compelling and novel. Thus, the paper is suitable for publication in The EMBO Journal. This Reviewer has only minor points, listed below, that can be helpful in improving the work:

We thank the reviewer for their kind words and are pleased that they find our conclusions are novel and interesting. Furthermore, we are contented to hear that the manuscript was pleasant to read and that our experiments, in this reviewer's opinion, are well executed.

1. In the introduction, when discussing dosage compensation mechanism, it might be worth citing PMID: 35575458

We thank the reviewer for the suggestion and we added this reference to our manuscript.

2. Also, when discussing proteotoxic stress, it might be worth citing PMID: 26404941

We have implemented this reference in our text.

3. When the authors describe the contribution of aneuploidy to heterogeneity, tumor progression, etc., it might be worth citing PMID: 34266888 and 34266887

We thank the reviewer for these suggested citations and have implemented them in our manuscript.

4. In the first paragraph of the results, it is worth adding that RPE1 cells are untransformed

We appreciate this suggestion and now describe the RPE-1 cell line used in this in much more detail. We make use of an RPE-1 cell line with a stable KD of p53. Besides this mutation, RPE-1 cells also harbor an oncogenic KRAS mutation as well as a missense mutation in CDKN2A (Libouban et al. 2017; Di Nicolantonio et al. 2008). TP53 loss in RPE-1 cells has been demonstrated to be sufficient to drive anchorage-independent growth, indicating that these cells display characteristics that are common to transformed cells (Chunduri et al. 2021). Moreover, dampening of the p53-dependent transcriptional response by HRAS^{G12V} in RPE-1 cells is sufficient to induce tumors in mice (Segeren et al. 2022). Therefore, we conclude that RPE-1 p53kd cells are likely not completely non-transformed as they display characteristics of early transformation. We elaborated more thoroughly on our model system in our revised manuscript in the introduction.

5. Also in the first paragraph of results, last sentence: "...all clones initially displayed a growth defect..." it would be more appropriate to use "proliferation defect" rather than "growth defect", unless the authors actually measured growth rather than proliferation. The same applies to the beginning of page 8 ("To evaluate if reducing cGAS/STING-mediated inflammation in early clones could rescue the slow growth phenotype").

We thank the reviewer for this suggestion and we changed "growth" in the manuscript to "proliferation", as this is indeed what we measured.

6. It looks like reference #6 and #74 are duplicated.

We apologize for this mistake and corrected it.

References

- Bakhoum, Samuel F., and Dan Avi Landau. 2017. "Chromosomal Instability as a Driver of Tumor Heterogeneity and Evolution." *Cold Spring Harbor Perspectives in Medicine* 7(6): a029611.
- Bao, Chunyang et al. 2023. "Genomic Signatures of Past and Present Chromosomal Instability in Barrett's Esophagus and Early Esophageal Adenocarcinoma." *Nature Communications* 14(1).
- Boekenkamp, Jan-Eric et al. 2024. "Proteogenomic Analysis Reveals Adaptive Strategies to Alleviate the Consequences of Aneuploidy in Cancer." *bioRxiv*: 2024.03.05.583460.
- Chunduri, Narendra Kumar et al. 2021. "Systems Approaches Identify the Consequences of Monosomy in Somatic Human Cells." *Nature communications* 12(1).
- Gao, Ruli et al. 2016. "Punctuated Copy Number Evolution and Clonal Stasis in Triple-Negative Breast Cancer." *Nature genetics* 48(10): 1119–30.
- Gerstung, Moritz et al. 2020. "The Evolutionary History of 2,658 Cancers." *Nature* 2020 578:7793 578(7793): 122–28.
- Knouse, Kristin A., Teresa Davoli, Stephen J. Elledge, and Angelika Amon. 2017. "Aneuploidy in Cancer: Seq-Ing Answers to Old Questions." <https://doi.org/10.1146/annurev-cancerbio-042616-072231> 1: 335–54.
- Krill-Burger, John M. et al. 2012. "Renal Cell Neoplasms Contain Shared Tumor Type-Specific Copy Number Variations." *The American Journal of Pathology* 180(6): 2427–39.
- Libouban, Marion A.A. et al. 2017. "Stable Aneuploid Tumors Cells Are More Sensitive to TTK Inhibition than Chromosomally Unstable Cell Lines." *Oncotarget* 8(24): 38309.
- Mitchell, Thomas J. et al. 2018. "Timing the Landmark Events in the Evolution of Clear Cell Renal Cell Cancer: TRACERx Renal." *Cell* 173(3): 611-623.e17.
- Di Nicolantonio, Federica et al. 2008. "Replacement of Normal with Mutant Alleles in the Genome of Normal Human Cells Unveils Mutation-Specific Drug Responses." *Proceedings of the National Academy of Sciences of the United States of America* 105(52): 20864–69.
- Ross-Innes, Caryn S. et al. 2015. "Whole-Genome Sequencing Provides New Insights into the Clonal Architecture of Barrett's Esophagus and Esophageal Adenocarcinoma." *Nature genetics* 47(9): 1038–46.
- Segeren, Hendrika A. et al. 2022. "Oncogenic RAS Sensitizes Cells to Drug-Induced Replication Stress via Transcriptional Silencing of P53." *Oncogene* 41(19): 2719.
- Teixeira, Vitor H. et al. 2019. "Deciphering the Genomic, Epigenomic, and Transcriptomic Landscapes of Pre-Invasive Lung Cancer Lesions." *Nature medicine* 25(3): 517–25.
- Wang, Yong et al. 2014. "Clonal Evolution in Breast Cancer Revealed by Single Nucleus Genome Sequencing." *Nature* 512(7513): 155.
- Yates, Lucy R., and Peter J. Campbell. 2012. "Evolution of the Cancer Genome." *Nature reviews. Genetics* 13(11): 795.

Dear Jonne,

Thank you for submitting your revised manuscript. It has now been seen by two of the original referees. I apologize for the delay in getting back to you. It took longer than anticipated to receive the referee reports given this busy time of the year.

As you can see, the referees find that the study is significantly improved during revision and recommend publication. However, I need you to address the points below before I can accept the manuscript.

- Please address the minor concerns of referee #1.
- During our routine analyses, we noted that there are textual overlaps with a document, which we realize is the PhD dissertation of the first author Dr. Dorine Hintzen (https://dspace.library.uu.nl/bitstream/handle/1874/452656/173709-dorine-hintzenprintversie-1_-_665844ad5cad4.pdf?sequence=1). Although EMBO Press policy does not disallow this, given that the dissertation is formally published, we would like to ask you to include formal citations to the PhD dissertations at all places where there are common blocks of text.
- Please rename the Conflict of interest section as "Disclosure Statement and Competing Interests".
- Please remove the Author Contributions section from the manuscript.
- As per our format requirements, in the reference list, citations should be listed in alphabetical order and then chronologically, with the authors' surnames and initials inverted; where there are more than 10 authors on a paper, 10 will be listed, followed by 'et al.'. Please see <https://www.embopress.org/page/journal/14693178/authorguide#referencesformat>
- We note that the ORCID ID of Dr. René Medema is currently not linked. As of January 2016, new EMBO Press policy asks for all corresponding authors to link to their ORCID iDs. You can read about the change under "Authorship Guidelines" in the Guide to Authors here: <https://www.embopress.org/page/journal/14693178/authorguide#authorshipguidelines>

In order to link your ORCID iD to your account in our manuscript tracking system, please do the following:

1. Click the 'Modify Profile' link at the bottom of your homepage in our system.
2. On the next page you will see a box halfway down the page titled ORCID*. Below this box is red text reading 'To Register/Link to ORCID, click here'. Please follow that link: you will be taken to ORCID where you can log in to your account (or create an account if you don't have one)
3. You will then be asked to authorise Wiley to access your ORCID information. Once you have approved the linking, you will be brought back to our manuscript system.

We regret that we cannot do this linking on your behalf for security reasons.

- We note the following regarding figure callouts: Table S1 called out but it is missing; the following callouts need to be correctly updated: "S5B", "S2D"
- Expanded View Table 1 is a Dataset and should be renamed and uplidd as such with the nomenclature Dataset EV1; the callouts in the manuscript text need to be updated accordingly.
- Please make the dataset PRJNA1119684 publicly available.
- Please provide GEO accession information on source data underlying Fig. 2A in the Data Availability section.
- Please upload source data as one file per figure.
- In the source data checklist, Figure 4F was marked as S4A, which we could not locate. Please provide more details on this.
- The manuscript sections should be in the following order: Title page - Abstract & Keywords - Introduction - Results - Discussion - Methods - Data Availability - Acknowledgments - Disclosure Statement & Competing Interests - References - Figure Legends - (Main Tables with legends) - Expanded View Figure Legends.
- Titles for EV figure legends and separate files need correction: it should be "Figure EV1" instead of "Expanded View Figure 1", etc.
- Materials and Methods should be renamed as Methods.
- Our production/data editors have asked you to clarify several points in the figure legends:
 - o We note that statistical analysis was performed on n=2 in figures 6C and 6E, which we discourage, or require justification for such a small n (please see <https://www.embopress.org/page/journal/14693178/authorguide#statisticalanalysis>).
 - o Please note that the figure 6c; EV 3a; does not contain any statistical parameters, kindly rectify the statistics test related information in the figure legend appropriately.
 - o Please note that the figure 1c is missing in the manuscript, although the legend for the same is provided in the manuscript. This needs to be rectified.
 - o Please define the annotated p values ***/**/* as well as provide the exact p-values for the same in the legend of figure 6e-g; EV 4a; as appropriate.
 - o Please indicate the statistical test used for data analysis in the legend of figure EV 5a.
- Papers published in EMBO Reports include a 'synopsis' and 'bullet points' to further enhance discoverability. Both are displayed on the html version of the paper and are freely accessible to all readers. The synopsis includes a short standfirst summarizing the study in 1 or 2 sentences (max 35 words) that summarize the paper and are provided by the authors and

streamlined by the handling editor. I would therefore ask you to include your synopsis blurb and 3-5 bullet points listing the key experimental findings.

- In addition, please provide an image for the synopsis. This image should provide a rapid overview of the question addressed in the study but still needs to be kept fairly modest since the image size cannot exceed 550 (width) x 300-600 (height) pixels.

Thank you again for giving us to consider your manuscript for EMBO Reports, I look forward to your minor revision.

Kind regards,

Deniz

--

Deniz Senyilmaz Tiebe, PhD
Senior Scientific Editor
EMBO Reports

Referee #1:

I want to thank the authors for incorporating my suggestions. This is a great manuscript, and I only have two minor comments:

- 1) Thank you for adding the statistics to Figure 2C! Could you also perform a paired t-test for Figure 2B?
- 2) Instead of referring to "CNV-seq", just say low pass whole genome sequencing. "CNV-seq" suggested a new method to me.

Referee #2:

The authors have satisfactorily addressed my prior points.

Also, they have carefully considered comments and suggestions from the other Reviewers and have provided convincing responses. Thus, I strongly support publication in EMBO Reports.

Referee #1:

I want to thank the authors for incorporating my suggestions. This is a great manuscript, and I only have two minor comments:

- 1) Thank you for adding the statistics to Figure 2C! Could you also perform a paired t-test for Figure 2B?
- 2) Instead of referring to "CNV-seq", just say low pass whole genome sequencing. "CNV-seq" suggested a new method to me.

Thank you for your kind words. We have implemented these last 2 points in the final revision.

Referee #2:

The authors have satisfactorily addressed my prior points. Also, they have carefully considered comments and suggestions from the other Reviewers and have provided convincing responses. Thus, I strongly support publication in EMBO Reports.

Thank you for your time and effort in reviewing our manuscript.

Dr. Jonne Raaijmakers
The Netherlands Cancer Institute
Department of Cell Biology
Plesmanlaan 121
Amsterdam 1066 CX
Netherlands

Dear Jonne,

Thank you for submitting your revised manuscript. I have now looked at everything and all is fine. Therefore, I am very pleased to accept your manuscript for publication in EMBO Reports.

Congratulations on a nice work!

Kind regards,

Deniz

--

Deniz Senyilmaz Tiebe, PhD
Senior Scientific Editor
EMBO Reports
